# Clonal selection confers distinct evolutionary trajectories in *BRAF*-driven cancers

Priyanka Gopal [1,2], Elif Irem Sarihan[1], Eui Kyu Chie [3], Gwendolyn Kuzmishin[1], Semihcan Doken[1], Nathan A. Pennell[4], Daniel P. Raymond[5], Sudish C. Murthy [5], Usman Ahmad[5], Siva Raja[5], Francisco Almeida[6], Sonali Sethi[6], Thomas R. Gildea[6], Craig D. Peacock[1], Drew J. Adams[7] & Mohamed E. Abazeed[1,8]*

Molecular determinants governing the evolution of tumor subclones toward phylogenetic branches or fixation remain unknown. Using sequencing data, we model the propagation and selection of clones expressing distinct categories of *BRAF* mutations to estimate their evolutionary trajectories. We show that strongly activating *BRAF* mutations demonstrate hard sweep dynamics, whereas mutations with less pronounced activation of the BRAF signaling pathway confer soft sweeps or are subclonal. We use clonal reconstructions to estimate the strength of "driver" selection in individual tumors. Using tumors cells and human-derived murine xenografts, we show that tumor sweep dynamics can significantly affect responses to targeted inhibitors of BRAF/MEK or DNA damaging agents. Our study uncovers patterns of distinct *BRAF* clonal evolutionary dynamics and nominates therapeutic strategies based on the identity of the *BRAF* mutation and its clonal composition.

[1] 2111 East 96th St/NE-6, Department of Translational Hematology Oncology Research, Cleveland Clinic, Cleveland, OH 44106, USA. [2] 2111 East 96th St/ND-46, Molecular Medicine Program, Lerner Research Institute, Case Western Reserve University, Cleveland, OH 44106, USA. [3] 101, Daehak-Ro, Jongno-Gu, Department of Radiation Oncology, Seoul National University College of Medicine, Seoul, Korea 110-774, USA. [4] 10201 Carnegie Ave/CA-5, Department of Hematology and Medical Oncology, Cleveland Clinic, Cleveland, OH 44195, USA. [5] 9500 Euclid Avenue/J4-1, Department of Thoracic and Cardiovascular Surgery, Cleveland Clinic, Cleveland, OH 44195, USA. [6] 9500 Euclid Avenue/M2-141, Department of Pulmonary Medicine, Cleveland Clinic, Cleveland, OH 44195, USA. [7] 2109 Adelbert Road/BRB, Case Western Reserve University, Department of Genetics, Cleveland, OH 44106, USA. [8] 10201 Carnegie Ave/CA-5, Department of Radiation Oncology, Cleveland Clinic, Cleveland, OH 44195, USA. *email: abazeem@ccf.org

nvestigations of several cancer types have led to the identification of genetic markers in critical genes that have successfully guided drug therapies[1–3]. However, cancers with specific genetic alterations do not respond uniformly to targeted therapies[4,5]. Enigmatically, differential responses can occur despite the presence of known activating mutations in the targeted driver. Ongoing clinical studies seek to map the sensitivity of tumors with individual molecular alterations across a variety of cancer types (e.g., multi-histology "basket" trials)[6,7]. However, it is increasingly evident that factors beyond the presence of the targeted molecular alteration can regulate tumor sensitivity. There is substantial clinical utility in the identification of these currently unknown tumor characteristics.

Evolutionary processes depend on the accumulation of genetic alterations to shape the clonal composition of cancer[8]. Tumor cells can establish subpopulations, or subclones, by acquiring new mutations that confer a fitness advantage, permitting relative expansion. Using high-coverage sequencing data, it is possible to infer the subclonal structure of a heterogeneous tumor using the population frequencies, or the variant-allele fractions (VAF), of the mutations that distinguish them[9–11]. The relative size of a subclone, and relatedly its selection, could be estimated from the average of its VAF cluster[12]. However, quantifying the magnitude of a subclone's fitness (e.g., the role of resident molecular drivers) and the influence of the strength of a subclone's selection on the global genetic composition of a tumor remains a challenge. Estimates of effect size across a population using the substitution rate of a variant can provide some indication of selection intensity, or the average selective effect[13]. However, this and other similar approaches do not quantify the strength of selection in individual tumors[14]. In cases in which the genetic drivers that regulate selection in an individual tumor can be targeted with a therapy, delineating the relationship between evolutionary processes and the probability of tumor extinction can prove decisive in guiding treatment strategies.

The identification of *BRAF* as a commonly mutated target in cancers has significantly altered the management of affected cancer types. *BRAF* is mutated in ~8% of all tumors including melanoma (~50%) or papillary thyroid (~60%), colorectal (~12%) or non-small cell lung cancer (~5%)[15–18]. By far, the most frequent mutation in *BRAF* across all cancer types is *BRAF*[V600E], which drives tumor growth by hyperactivating the mitogen-activated protein kinase (MAPK) signaling pathway. In addition to *BRAF*[V600E] there are many other mutations in exons that encode or are directly adjacent to the conserved kinase domain[19]. Some of these mutations have been previously characterized as hypermorphic, but a majority remain categorized as variants of unknown clinical significance. For example, in lung adenocarcinoma, it is estimated that approximately half of the mutations in *BRAF* are non-V600[20]. Despite the confirmed activation of the mitogen-activated extracellular-signal-regulated kinase (MEK/ERK) pathway in some of these tumors, it is not clear whether mutations in these putative *BRAF*-driven tumors confer upfront sensitivity to inhibitors of BRAF (BRAFi) and/or MEK (MEKi) or alter the tumor genetic composition[21–23]. Due to its multitude of variants, its significant alteration frequency in several cancer types and the variable clinical efficacy of drugs that target it, *BRAF* is a model oncogene to study molecular classification-based heterogeneity.

Herein, we map the clonal trajectory of *BRAF* mutation-bearing cells across diverse variants and cancer types and exploit the variation in the architectures of *BRAF*-driven tumors to optimize tumor sensitivity.

## Results

**Phenotypic impact profiling of *BRAF* variants.** We identified 405 candidate *BRAF* variants by analyzing targeted and genome-wide screen data from a collection of 48,397 tumors representing 35 cancers deposited in COSMIC[19]. As expected, most tumors contained a *BRAF*[V600E] mutation (Fig. 1a). Variants were more likely to be found in multiple cancer types as their frequency across the population increased. The majority of the variants were infrequently found in human cancers (Fig. 1a). Specifically, 306 (74.5%) of the variants were found only once. Several other residues had modest variant frequencies including G464, G466, G469, N581, D594, and L597 (Fig. 1b). These residues comprise the activation loop (A-loop) near V600 (L597), the phosphate binding loop (P-loop) (464–469), and residues critical for chelation of the divalent $Mg^{+2}$ associated with ATP to help orient the molecule for optimal substitution (D594 and N581) (Fig. 1c). The top 10 most common variants showed cancer type preferences (Fig. 1d). The vast majority of the non-V600 variants identified were variants of unknown significance.

We reasoned that a comparison of gene expression changes induced by variants of *BRAF* could provide functional insight into their phenotypic impact. We selected seven variants considering local mutational density, evolutionary conservation and ontological curation (Supplementary Fig. 1) and 28 *BRAF* variants by random sampling. We used site-directed mutagenesis to generate mutant clones and transferred alleles into lentiviral vectors. Overall, we generated 74 expression constructs including wild-type and vector controls and experimental replicates. We then stably-expressed each variant in transformed bronchial epithelial cells (BEAS-2B).

Total mRNA gene expression was assayed using RNAseq. A gene signature composed of the most variable genes was selected to estimate BRAF activity (Supplementary Fig. 2). BRAF scores across all variants had wide and graded variance (Fig. 1e). Importantly, V600E had a high score and the previously characterized low-activity mutation G466V had a low score[24]. Genes that comprised the BRAF score significantly overlapped with gene sets that measure epithelial-to-mesenchymal transition ($P = 9.05 \times 10^{-26}$; hypergeometric test and $Q = 2.26 \times 10^{-26}$; corrected false discovery rate) and *KRAS* signaling ($P = 6.11 \times 10^{-11}$; hypergeometric test and $Q = 7.64 \times 10^{-10}$; corrected false discovery rate), consistent with relevant BRAF-related biological pathways (Supplementary Table 1). Moreover, the BRAF score was highly correlated with ERK pathway activity (Pearson $r = 0.783$; Supplementary Fig. 3). To determine the concordance between gene expression changes and the phenotypic impact of individual variants, we characterized the in vivo tumorigenicity of a subset of variants. Variants with high BRAF scores formed tumors rapidly (within 4 weeks) and tumors with low signature scores took significantly longer (2–3 months) or did not engraft (Fig. 1f). The BRAF score were inversely proportional to the time to $1 \text{ cm}^3$ (Fig. 1f, inset), indicating an association between tumor doubling times and BRAF activity.

**BRAF-variant fitness and tumor clonal architecture.** We posited that the fitness of a particular *BRAF*-variant will determine the host tumor's clonal architecture. We modeled the impact of the fitness of the *BRAF*-variant-containing subclones and the variant acquisition times on the tumor fraction using:

$$x(t) = \frac{e^{st}}{e^t + e^{st}} \qquad (1)$$

where *s* is the relative rates of growth between the *BRAF*-variant subclones compared to the (wild-type) host tumor[25]. The relative growth rates were estimated from categories of (low, medium and high selection) variants in Fig. 1e. We defined *t_sc*, which was measured in generations, as the time that a subclone arose. Given an estimate of the age of a tumor, simulations of varying *s* and *t_sc* demonstrated that variants with high fitness were significantly

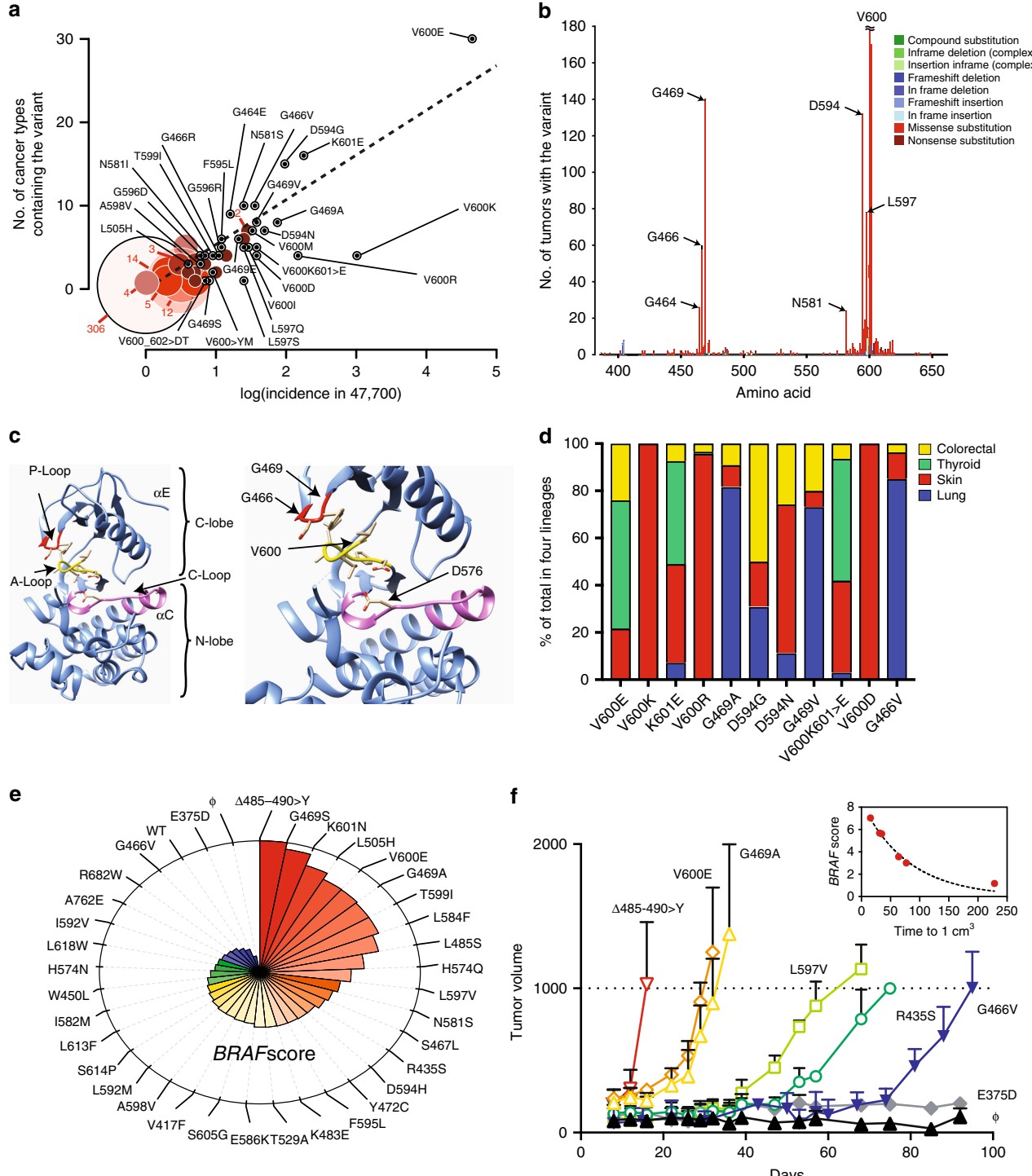

**Fig. 1** The landscape of genetic drivers in *BRAF* features distinct variants. **a** The size of the circle corresponds to the number of variants at that amino acid position. Variants that occupied a unique position are annotated. The vast majority of *BRAF* variants were found once in a single cancer type. **b, c** Secondary (non-V600) variant frequency peaks occur in residues that comprise the A-loop, the P-loop and residues critical for $Mg^{+2}$ chelation. The catalytic D576 (C-loop), which is in a cleft between the N- and C-lobes, is shown. **d** The relative proportion of the 10 most frequent variants in the four most common cancer types are shown. **e** Clock plot of *BRAF* signature score in BEAS-2B cells expressing vector control (φ), wild-type (WT) or 35 BRAF variants. Red and blue represents cells with the most and least BRAF activity, respectively. **f** BEAS-2B cells stably infected with vector alone (φ) or vector expressing *BRAF* alleles were injected into the flanks of NSG mice and monitored for growth. The association between the *BRAF* signature score and the time for tumor volume to reach 1 cm³ is shown in the inset. Tumor volume is expressed as the mean ± s.d. of at least six independent biological replicates

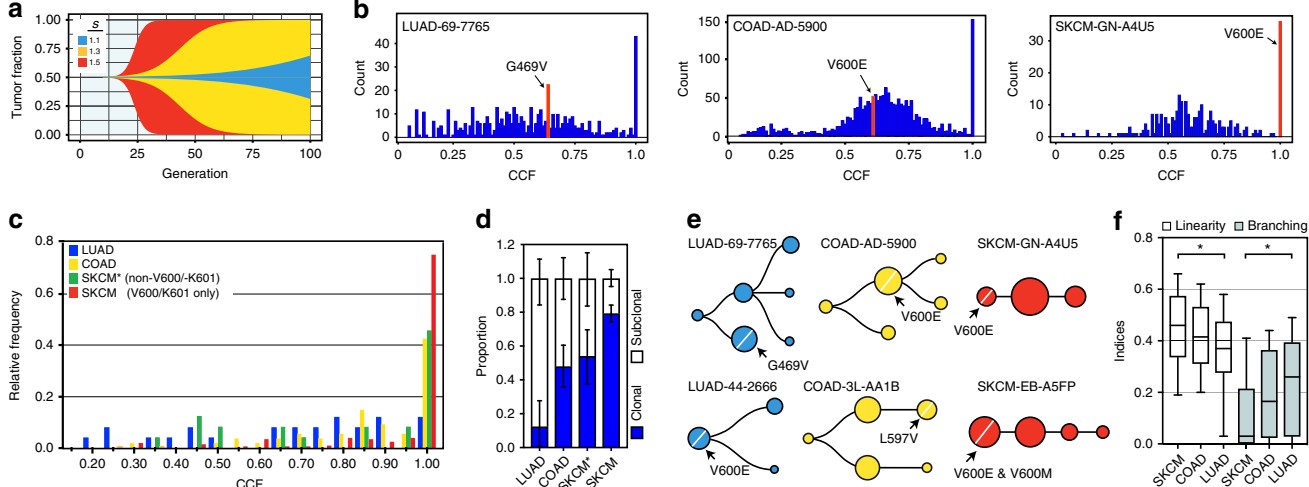

**Fig. 2** Measuring *BRAF*-variant fitness in sequencing data. **a** Subclones with low or high fitness advantages achieve fixation at distinct rates when under selection. Tumor and subclone ages were 100 and five generations, respectively. **b** Allelic fractions for mutations in three representative tumors from TCGA were rescaled to estimates of cancer cell fraction (CCF) by correcting for sample purity and local copy-number. The position of the *BRAF*-variant CCF is designated in red. **c** The relative frequency of *BRAF*-variant CCF values across cancer type is shown. SKCM was further stratified by the hyperactivating mutations V600 and K601. **d** Proportion of tumors with estimated subclonal *BRAF*-drivers across cancer types. **e** The clonal evolutionary structure of each tumor is depicted in the form of a rooted tree. The tree with the lowest normalized *log* likelihood value ("best tree")") is shown for representative tumors from each cancer type. Cancer types are organized by color: LUAD (blue), COAD (yellow) and SKCM (red). The root node represents the clonal fraction and branched nodes represent subclones. Node size reflects the number of mutations that constitute the (sub)clone. Arrows indicate the position of the *BRAF* variant(s) in the tree. **f** Box-plot (median, inter-quartile range, and minimum/maximum) of linearity and branched indices for evaluable tumors across the designated cancer types are shown. Only primary (non-metastatic) tumors were evaluated. The *P*-value of Welch's *t*-test comparing SKCM to LUAD was <0.05 and <0.01 for linearity and branching, respectively

more likely to achieve fixation, which is defined as present in all tumor cells, compared to less fit variant-containing subclones (Fig. 2a)[26].

Based on this model, we reasoned that rarer non-V600 *BRAF* variants would either fixate slowly or remain confined to a phylogenetic branch. We tested this prediction using data from TCGA. First, we combined gene-level copy-number and mutational data with estimates of the sample's purity (fraction of tumor cells) to infer the cancer cell fraction (CCF), or the proportion of cancer cells with the variant (Fig. 2b). We estimated the clonality of *BRAF* variants in 22 (3 V600, 19 non-V600), 44 (33 V600; 11 non-V600), and 187 (156 V600, 21 non-V600) lung adenocarcinoma (LUAD), colorectal adenocarcinoma (COAD), and melanoma skin cancer (SKCM) tumors, respectively. We found that the CCF of the *BRAF* variants varied significantly on the basis of cancer type, with ~9% of analyzed LUAD containing *BRAF* variants that were clonal compared to >80% of SKCM (Fig. 2c, d). Estimates of the ratios of clonal BRAF V600E variants in LUAD, COAD and SKCM tumors were 0.33, 0.52, and 0.76, respectively. Importantly, SKCM tumors derived from sites of metastases (74% of tumors analyzed) did not demonstrate significant differences in the frequency of *BRAF* CCF values compared to tumors sampled from primary sites (Supplementary Fig. 4). This observation suggested that fixation of *BRAF* variants in SKCM preceded metastatic spread.

We reconstructed the global clonal composition of genotypes of the *BRAF*-driven tumors[27]. We found that phylogenetic trees of LUAD and COAD tumors displayed evidence of significant branching, whereas SKCM tumors had a mainly linear trajectory (Fig. 2e, f). The latter model suggests that *BRAF*[V600E]-driven SKCM tumors have a major dominant clone with only rare intermediates from previous sweeps. Moreover, consistent with our CCF estimates, *BRAF* variants were significantly more likely to be clonal in SKCM compared to COAD or LUAD. Altogether, these data demonstrated distinct clonal compositions in *BRAF*-

driven tumors across cancer types and the propensity for more linear evolutionary trajectories in *BRAF*[V600E]-driven SKCM tumors.

**Copy gains occur preferentially at the *BRAF*-variant locus.** Patients with *BRAF*[V600E] SKCM have an impressive overall response rate (ORR) of 64–87% after combined BRAFi/MEKi[28–30]. Tumor regression upon initiation of therapy can be so dramatic that the phenomenon has been described colloquially as the 'Lazarus syndrome'[31,32]. Although there are reports of clinical responses in patients with *BRAF*[V600E] containing LUAD[5,33] and COAD[34], these are typically lower and less dramatic than those observed in SKCM. Our estimates of clonality suggested that *BRAF*[V600E]-mutated cancers appear to have distinct clonal compositions based on the cancer type of origin. We sought a genetic basis for the differences in architecture by first identifying co-occurring genetic alterations with *BRAF* mutations in SKCM. We generated covariate associations with *BRAF* mutations in 389 SKCM samples using 1489 genetic features and found a strong association with focal amplification at 7q34 ($P = 1.4 \times 10^{-7}$; pairwise Fisher's exact test) and arm-level amplifications of 7q and 7p (Supplementary Data 1 and Fig. 3a).

Since 7q34 includes the *BRAF* locus, we examined the association between *BRAF* gene-level gain or amplification, gene expression and mutation more directly. SKCM mutated *BRAF* were much more likely than LUAD or COAD to have focal *BRAF* copy-number gains (Fig. 3b, c). Copy gains were associated with higher *BRAF* gene expression (Fig. 3b) and protein levels (Supplementary Fig. 5). Of the mutant *BRAF* SKCM, ~80% had gene-level *BRAF* gain or amplification (Fig. 3c). The patterns of somatic copy-number alterations across the three cancers showed that only SKCM tumors had significantly more focal amplifications (7q34) targeting *BRAF*, further supporting this association (Fig. 3d). Importantly, an inclusion criterion for the samples

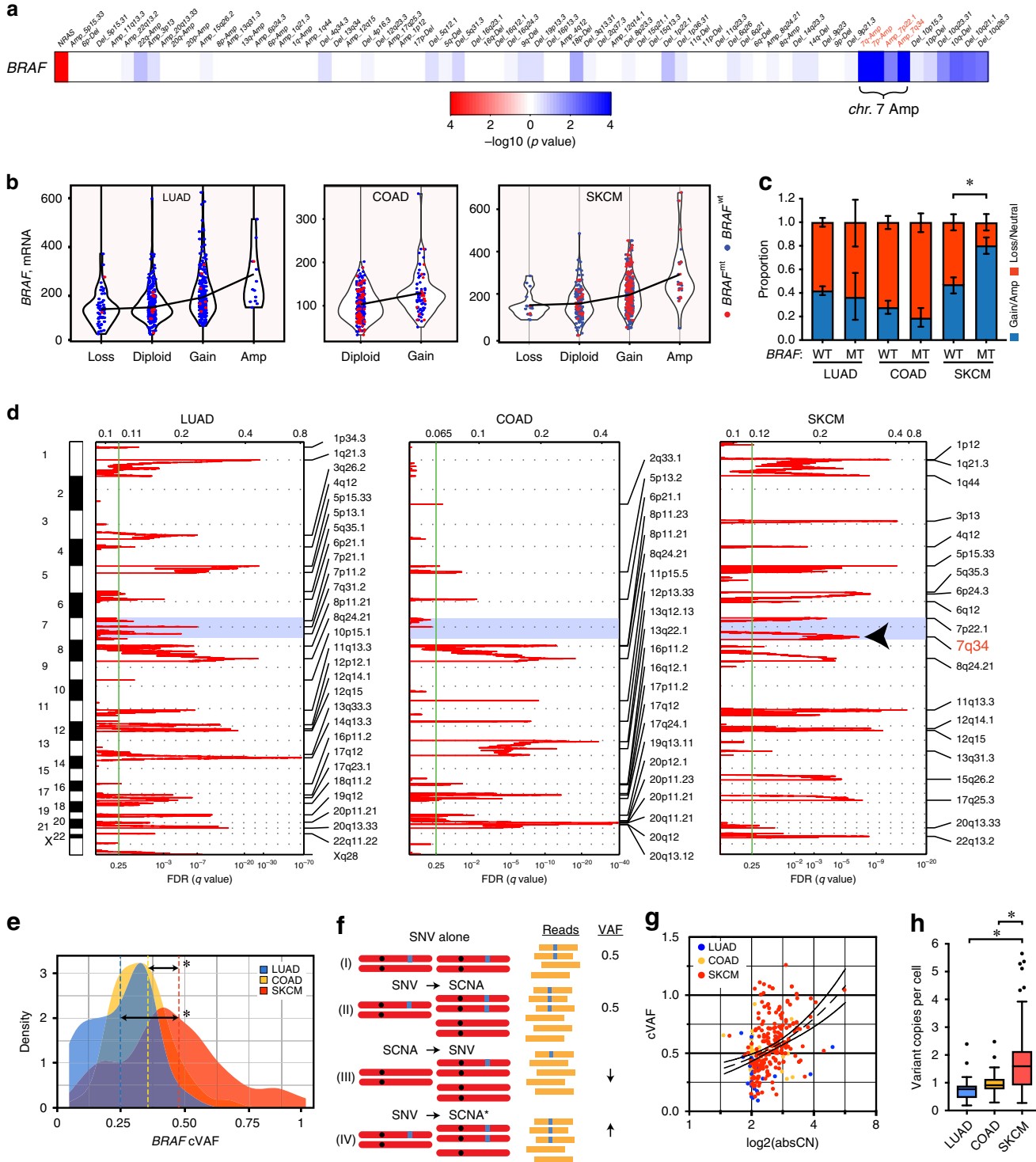

profiled by TCGA was no previous systemic therapy (except adjuvant interferon-α > 90 days prior), indicating that *7q34* amplification is not a consequence of acquired resistance to BRAFi/MEKi in these tumors[35,36].

The co-occurrence of *BRAF* mutations and copy-number gains suggested cooperation between the two alterations. Such an interaction could be functionally significant if the focal gain occurs at the variant *BRAF* locus. We used the VAF estimate to determine the phylogenetic order of mutation and copy gain and the allele preference for gain (wild-type or mutant). First, we calculated the density distribution of the VAF of each cancer type

corrected for purity (and not ploidy) or cVAF (Fig. 3e). The cVAF was significantly higher for SKCM, with many samples exceeding a value of 0.5. The orders between gene copy gain and mutation is predicted to differentially impact the VAF at an individual locus (Fig. 3f). Of the several permuted orders of events, the *BRAF* VAF will only exceed 0.5 if the *BRAF* mutation precedes the gain and there is a preference for the gain on the variant locus. Consistent with this order of alterations, we show that the cVAF is directly proportional to the absolute *BRAF* copy-number (Fig. 3g) and the *BRAF* cellular multiplicity, or the number of variant copies per cell, was significantly higher in

**Fig. 3** *BRAF* mutation is frequently followed by variant-selective amplification in SKCM. **a** Co-occurring (blue) and mutually exclusive (red) copy number and other mutation events with *BRAF* mutations. *P*-values were calculated using the pairwise Fisher's exact test. **b** Violin plot of *BRAF* mRNA organized by putative copy number alteration frequency estimated by GISTIC. The horizontal line connects median values of mRNA expression in each group. **c** The proportion of tumors with copy gains organized by *BRAF* genotype and cancer type. The *P*-value of the binomial test was <0.05. Confidence intervals were calculated by the Clopper and Pearson exact test. **d** GISTIC analysis of copy-number changes in each cancer type. FDR *Q* values account for multiple-hypothesis testing. The significance threshold is indicated by the green line. The locations of the peak regions are indicated to the right of each panel. Chromosome positions are indicated along the *y*-axis with centromere positions indicated by dotted lines. The blue band delimits chromosome 7. The arrowhead indicates the position of focal amplification at 7q34 is SKCM. **e** Probability density function of *BRAF* VAF in LUAD, COAD, or SKCM. The mean is indicated in dashed line. The *P*-value of Welch's *t*-test comparing the mean of SKCM to LUAD or COAD were <0.0001. **f** The dependence of VAF on the phylogenetic relationship between a locus-specific somatic copy-number gain (SCNA) and single-nucleotide variant (SNV). **g** Scatter plot and linear regression (dashed line) of cVAF and absolute copy-number stratified by cancer type. The slope is non-zero (*P* < 0.0001). **h** Allelic fractions were reinterpreted as average variant copies per cancer cell (or multiplicity). Box-plots show the median, the inter-quartile range, and the minimum/maximum after excluding potential outliers. The *P*-value of Welch's *t*-test comparing the means of SKCM to LUAD or COAD were <0.0001

SKCM (Fig. 3h). Altogether, these results suggest that *BRAF* mutations precede copy gain, and the latter preferentially occurs at the *BRAF*-variant locus in SKCM.

**Gain at the *BRAF*-variant locus causes hard selective sweeps.** Mutations in major cancer driving oncogenes are often mutually exclusive, especially if the oncogenes participate in the same pathway[37]. *BRAF*[V600E] mutations and copy-number gains in SKCM, however, appears to be an exception, which we hypothesized is due to functional cooperation between the two alterations. To test this, we induced the expression of *BRAF* variants V600E, G469A and G466V in BEAS-2B cells engineered to stably express each allele under a doxycycline (Dox)-inducible promoter. V600E and G469A had a broad dynamic range of pERK and pMEK activity (Supplementary Fig. 6). However, increasing the expression of variant G466V did not substantially impact pathway activity, suggesting that copy-number gains of this variant is not likely to confer a significant increase in cellular fitness. We also measured the in vivo growth dynamics of BEAS-2B cells expressing *BRAF*[V600E] driven by the human phosphoglycerate kinase 1 (PGK) or the elongation factor-1α (EF-1α) promoters. The higher variant level expression induced by the latter lead to a concomitant increase in BRAF-pathway activity and a significant increase in the rate of tumor growth. Specifically, a 5.2-fold difference in BRAF expression resulted in a fitness benefit of 1.42 (ratio of the slopes of growth of EF-1α to PGK driven tumors) (Supplementary Fig. 7). These results demonstrated that variant levels can significantly increase the activity of hyperactivating *BRAF* mutations and promote faster growth.

A consequence of the phylogenetic order and functional cooperation between *BRAF* mutation and copy-number gains in SKCM is the anticipated rapid expansion and fixation of the affected clone due to positive selection, or a selective sweep. We sought hallmark signatures in the tumor genetic data that indicate recent adaptation in the affected tumors, including a decrement in genetic diversity[38]. Specifically, in a rapid or 'hard' selective sweep, the expanding clone is expected to collapse all lineages into a single cluster, reduce genetic diversity and increase the frequency of alleles associated with the driver (or 'hitch-hikers') (Fig. 4a). We applied principles from estimates of haplotype frequencies in a population to estimate the strength of cancer cell clonal selection by measuring the frequency of post-selection passenger alleles or new hitch-hikers[39]. First, we modeled the frequency trajectory of a new adaptive mutation using logistic growth as follows:

$$n(t) = \frac{e^{st}}{e^{st} + 2Ns} \qquad (2)$$

where *s* is the fitness, *t* is the time measured in units of generations, *N* is a population of constant size, and $n \approx (2Ns)^{-1}$ is

the population frequency in which a fit variant is established. Although the model assumes a constant *N*, most tumors have a growth fraction that can be, in part, balanced by cell loss. Population size changes can make it difficult to distinguish selection from demographic processes, but only in cases in which there is weak selection[40]. Importantly, once a variant is established, it escapes stochastic loss and can be modeled by logistic growth[41]. To model the frequency trajectory of the *i*th passenger mutation we used:

$$n_i(t) = e^{-\mu t}\left(\frac{\mu}{is}\right)^{1-\mu/s} \qquad (3)$$

where $i \geq 1$, $\mu$ is the rate at which neutral mutations occur on the sweeping clone and *s* is equal to the fitness of the adaptive subclone (Fig. 4b and Supplementary Software 1).

We posited that if selection causes fixation rapidly, a decrement in the low-frequency range of the CCF distribution would be observed, resulting in lower genetic diversity. However, if the variant sweeps more gradually, a greater number of low-frequency neutral mutations could accumulate. There were characteristic signatures in the CCF density distributions that appeared to distinguish SKCM tumors with *BRAF* mutations from other tumor types; namely, the lack of low-frequency passenger variants (Supplementary Fig. 8). To quantitate these differences, we first calculated the median CCF of tumors with *BRAF* mutations across all three cancer types. The median CCF was significantly associated with *BRAF*-variant multiplicity, reflecting a shift in the overall allelic fraction of the tumor toward higher CCF values (non-neutral alleles) (Fig. 4c). In addition, we estimated genetic diversity of each tumor by measuring the information entropy (see *Methods*). There was an association between *BRAF*-variant multiplicity and decreased genetic diversity (Fig. 4d). These results indicated a reduction in the genetic diversity around *BRAF* subclones with high variant copy levels and corroborated the linear evolutionary trajectories predicted by phylogenetic estimates in these tumors.

Importantly, we accounted for several factors that could influence the signatures of CCF in tumors with high *BRAF*-variant multiplicity independent of clonal sweep dynamics. First, SKCM have the highest somatic mutation frequency of any tumor profiled to date[18], suggesting that the low-frequency variant differences in SKCM are not attributed to the overall neutral mutation rate in this tumor type. Second, sequencing coverage in SKCM TCGA had a > 80% power to detect subclonal mutations at 6% allelic fraction, indicating that the lack of mutations in this range of the distribution is not attributed to differential sequence coverage compared to LUAD and COAD[18]. Lastly, *BRAF*-driven SKCM harvested as primary tumors, which represented 26% of tumors profiled by TCGA, did not demonstrate significant differences in the frequency of clonality (Supplementary Fig. 4)

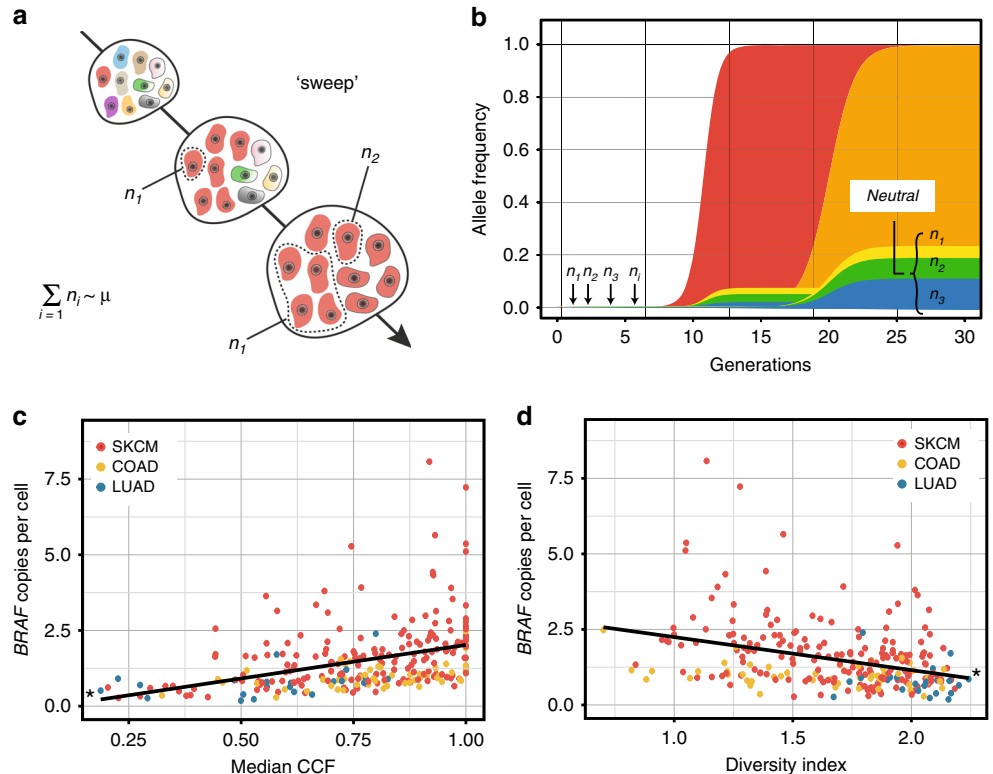

**Fig. 4** *BRAF*-variant multiplicity regulates subclonal dynamics. **a** Schematic of a selective sweep. The subclone under selection is in red. Progression along the arrow represents evolving clonal architecture. The accumulation of the *i*th passenger alleles associated with the subclone during selection (or hitch-hikers) is shown. μ is the neutral mutation rate. There is decreased genetic diversity as a result of the yielding of tumor subclones to a single clonal cluster if there is a rapid or hard selective sweep. **b** Modeling of the frequency trajectory of a new adaptive and *i*th passenger mutation during a selective sweep. Sweep parameters are $s = 2.0$ (hard) or 1.1 (soft) and μ = 0.001. The linear associations between the *BRAF*-variant multiplicity and the median **c** CCF or **d** genetic diversity across cancer types are shown. The slopes are non-zero ($P < 0.0001$)

or in genetic diversity (Supplementary Fig. 9) compared to metastases. Altogether, these data suggest that hard sweeps occur in *BRAF*-driven SKCM tumors and appear to precede metastasis.

**BRAF-variant multiplicity predicts response to BRAFi/MEKi.** Allelic signatures suggested that *BRAF* mutation and subsequent copy-number gain can drive affected subclones to fixation, leading to significant reduction in the genetic diversity of some tumors. We hypothesized that a population collapse around a single subclone may render a tumor more vulnerable to extinction when treated with BRAF-pathway inhibitors. We examined the effects of introducing BRAFi or MEKi in tumor cells with varying strengths of *BRAF*-variant selection. We calculated the variant multiplicity of 55 *BRAF*-mutated LUAD, COAD, and SKCM cell lines. We combined multiplicity values with BRAFi or MEKi sensitivity measurements derived from a recent large-scale drug sensitivity profiling effort[42]. *BRAF* copies per cell were significantly associated with an improved response to BRAFi or MEKi (Fig. 5a). We re-tested drug response rates in a subset of the LUAD cell lines using a growth delay experiment and confirmed the association between non-V600 *BRAF* mutant cells and MEKi-induced growth inhibition (Supplementary Fig. 10). Moreover, patient-derived xenografts (PDX) containing non-V600 (subclonal), V600 (clonal) or V600 with amplification (clonal) were treated with combined BRAFi/MEKi and showed progressive disease, stable disease and complete response, respectively (Fig. 5b; Supplementary Table 2). The extent of tumor volume changes correlated with estimates of *BRAF* copies per cell. Lastly, we stratified patients with SKCM using data from TCGA on the basis of *BRAF* mutation status and mRNA

expression. Patients with tumors containing *BRAF* mutations and high *BRAF* mRNA had improved overall survival compared to other cohorts ($P = 0.045$; *log*-rank trend; Supplementary Fig. 11). Together, these data indicated that *BRAF* cellular multiplicity affects BRAFi/MEKi sensitivity and could potentially result in improved patient outcomes.

**Optimal BRAF-variant gene dose and response to BRAFi/MEKi.** Resistance to BRAFi, MEKi and ERK kinase inhibitors (ERKi) has been shown to be highly correlated with the emergence of copy-number gains at the mutated *BRAF* gene locus[35]. We posited that a non-linear relationship between gene dosage and fitness under selection may explain the association between *BRAF* gene amplification, tumor cell fitness and therapeutic resistance to BRAF therapies. We plated, in equal proportion, BEAS-2B cells stably expressing *BRAF*[V600E] or *BRAF*[G466V] and vector alone under a Dox-inducible promoter (Fig. 5c). We induced the expression of V600E or G466V and assessed the cellular sensitivity to BRAFi or MEKi after 5 days of treatment. For V600E, we observed an initial decrement in survival (negative slope) at lower doses of Dox followed by resistance (positive slope) at higher doses (Fig. 5d). These results indicated a non-linear, convex relationship between BRAF activity and response to targeted inhibitors. It also suggested there is an optimal range of BRAF activity for enhanced cellular fitness that can be modulated in response to BRAFi. In contrast, higher levels of Dox were necessary to cause sensitivity to MEKi for G466V and resistance was not observed in Dox doses up to 100 ng/mL, consistent with the limited dynamic range in MEK/ERK activity despite increasing levels of G466V described earlier (Fig. 5e;

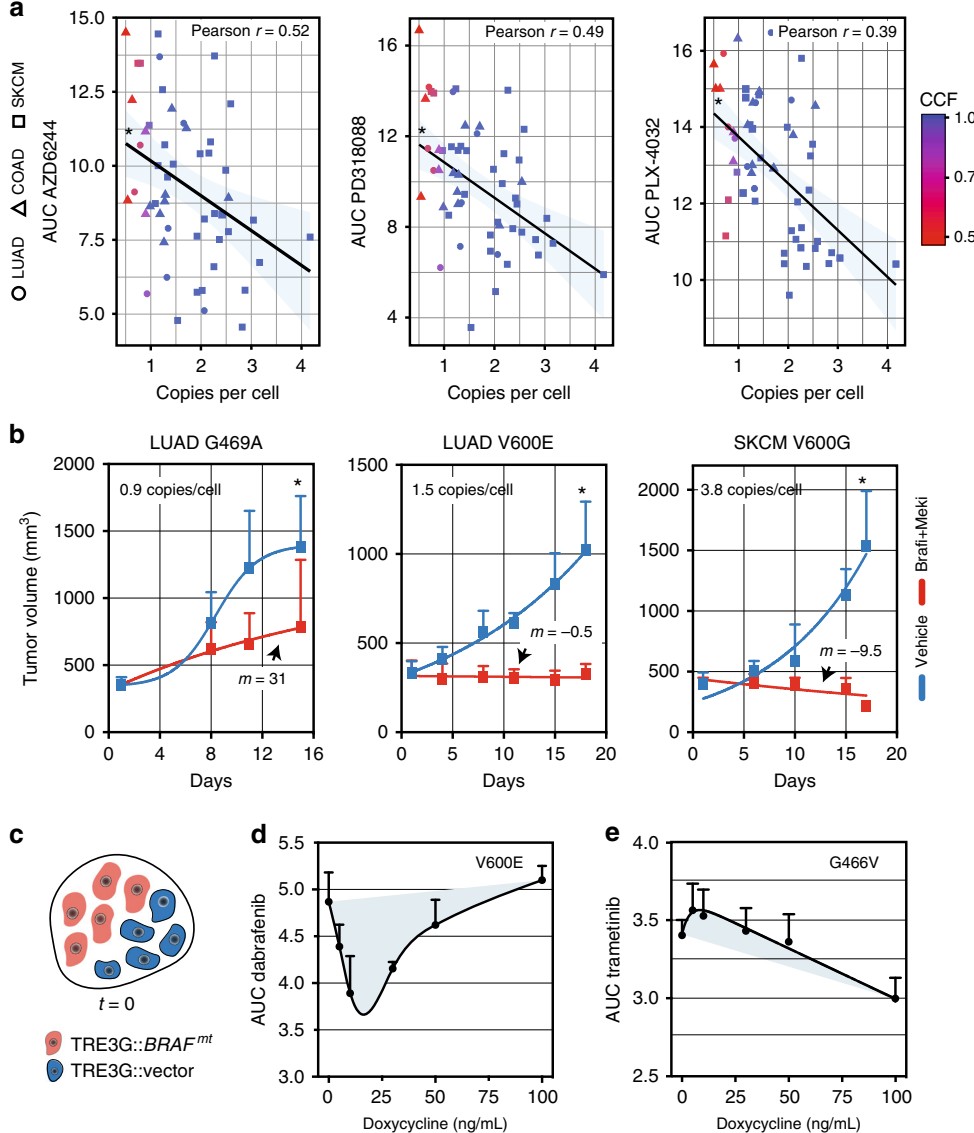

**Fig. 5** *BRAF*-variant identity and multiplicity regulates tumor responses to targeted therapies. **a** Cells with higher *BRAF* variants copies per cell (and higher CCF) were more sensitive to MEK1/2 (AZD6244 or PD318088) or BRAF (PLX-4032) inhibition. AUC, area under the curve. CCF values for individual cells are designated by the heatmap. **b** NSG mice bearing PDX with the indicated *BRAF* mutation and copies per cell were treated with dabrafenib (BRAFi) and trametinib (MEKi). Data are expressed as the mean ± s.d. The *P*-value of the χ2-test between the control and treatment groups was <0.05, <0.001, <0.0001. **c** Schematic depicting the co-culturing of cells expressing *BRAF*^V600E or *BRAF*^G466V and vector alone. After 48 of Dox-induction, cells were treated with either **d** dabrafenib (BRAFi) or **e** trametinib (MEKi). Cellular survival was measured 5 days after drug treatment. Data are expressed as the area under the curve (AUC) and represent the mean ± s.e.m. of at least three independent experiments

Supplementary Fig. 6). G469A, a high-activity variant, and R435S, a low-activity variant, demonstrated relationships between variant gene dose and drug response similar to V600E and G466V, respectively (Supplementary Fig. 12). These results suggest that both variant identity and dose regulates clonal sweep dynamics, and relatedly, response to targeted agents.

***BRAF* gain-of-function variants promote survival after DNA damage**. Since tumors with a soft sweep trajectory had mainly partial responses to BRAFi/MEKi, we tested combinatorial therapeutic strategies to improve treatment responses. First, we queried the role of these variants on the sensitivity of tumors to DNA damaging agents. We used a previously benchmarked high-throughput profiling method to study the effects of DNA damage on the survival of LUAD cells to ionizing radiation[43,44]. Genomic

correlates of radiosensitivity were calculated using a rescaled mutual information metric, the information coefficient (IC), a non-linear correlation coefficient that takes values between 1 (perfect association) and 0 (no association) (Fig. 6a).

Correlation with cancer genomic data in 28 LUAD cell lines identified *BRAF* mutations located in or near the highly conserved kinase domain as strongly associated with resistance to DNA damage. All functional mutations were confirmed to be located in or near the highly conserved kinase domain, with at least one mutation (R435S) not having been previously characterized as a functionally significant variant (Fig. 6b). All of the mutations associated with resistance had enhanced kinase activity. We expressed each of the identified variants and *BRAF*^V600E in a genetically defined, immortalized human bronchial epithelial cell line (BEAS-2B). We showed that non-V600 *BRAF* variants conferred resistance to ionizing radiation

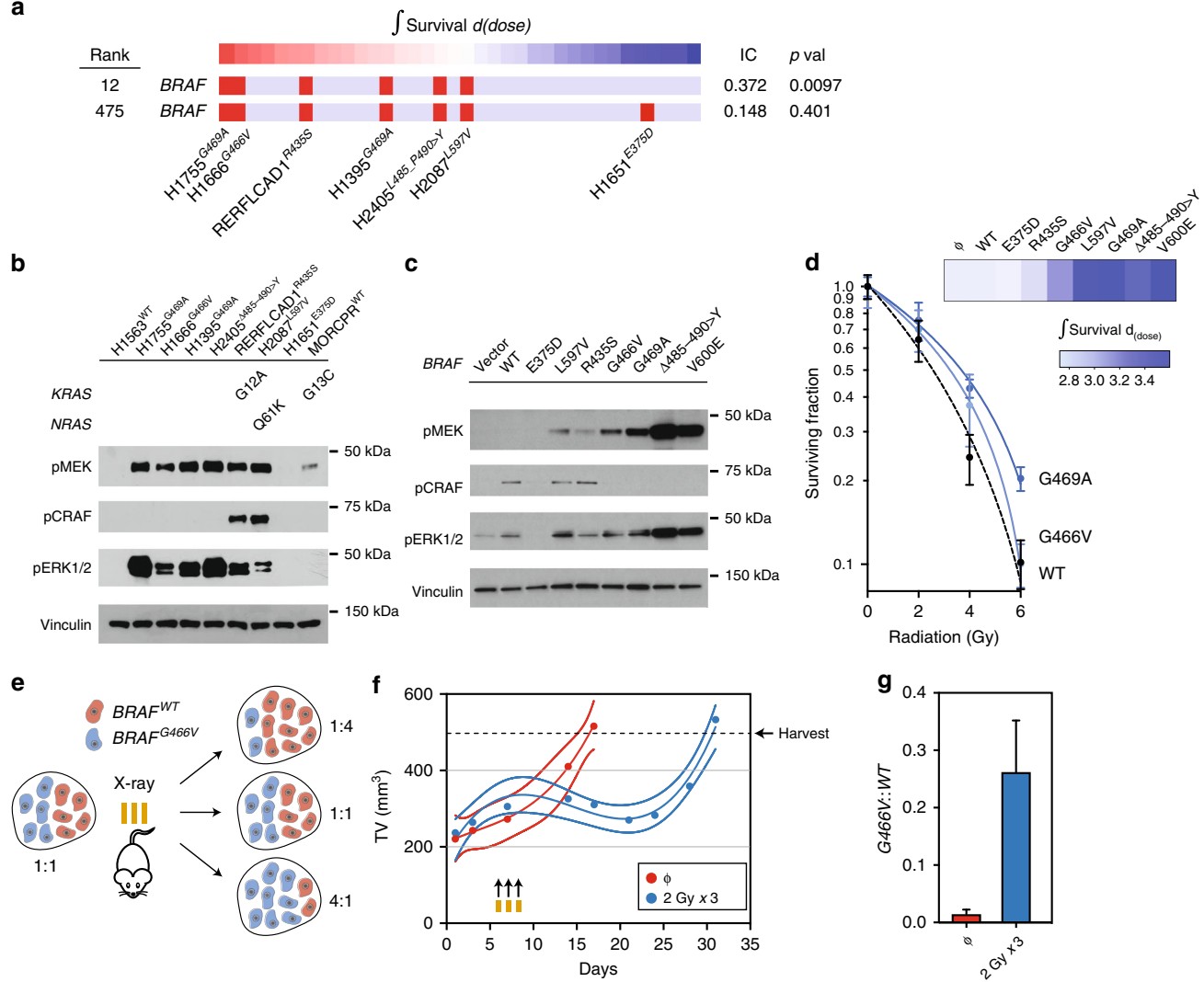

**Fig. 6** Hypermorphic *BRAF* variants confer resistance to genotoxic stress in LUAD. **a** Association between integral survival and *BRAF* genotype in 28 LUAD cell lines. Red bar represents mutation. *BRAF* mutation ranked 12 out of 6743 genomic features after outlier exclusion. The *P*-values were calculated using the empirical permutation test. **b** Immunoblot analysis of representative cell lines profiled for radiation sensitivity. **c** BEAS-2B cells stably infected with vector alone (ϕ) or vector expressing *BRAF* variants were profiled for RAF-MEK-ERK pathway activity by immunoblot. **d** BEAS-2B cells in **c** were treated with ionizing radiation and cell number was determined on days 7–9. Representative curves are shown. Data points represent mean ± s.e.m. The heatmap of integral survival is organized by the order of all of the transduced cells in **c**. **e** Schematic depiction of the experimental design used in **f**. **f** BEAS-2B cells expressing *BRAF* G466V or wild-type (WT) were injected into the flank of NSG mice and block randomized into each treatment arm: Mock (ϕ) or X-ray (2 Gy × 3). Tumor volumes were measured at least twice weekly. Data represent the mean. Solid line represents the interpolation of mean using a third order polynomial fit. Dashed lines represent the 95% confidence interval of the polynomial fit. *n* = 5 independent animals for each condition. **g** The ratio of G466V to WT was determined in the harvested tumors (at a volume of ~500 mm³) using ddPCR. The proportion in each arm was normalized to the fractional abundance in cells expressing G466V alone. Data are expressed as the mean ± s.e.m. of three independent experiments

and that the extent of resistance correlated with MEK/ERK pathway activation (Fig. 6c, d). These results indicated that most hypermorphic *BRAF* variants confer resistance to DNA damaging agents.

To test the model of *BRAF*-variant-containing subclone expansion during therapeutic stress, we examined the effects of a low-activity variant (*BRAF*^G466V) on subclone composition. We injected BEAS2B cells expressing *BRAF*^wt or *BRAF*^G466V in equal proportion into the flanks of NSG mice (Fig. 6e). Mice received either sham or actual irradiation and were allowed to recover to a size of 500 mm³; harvest volumes were similar in both arms (Fig. 6f). We then examined the relative proportion of the G466V variant compared to wild-type with or without irradiation. We observed a significant increase in the allelic fraction of G466V in

the irradiated mice, suggesting the preferential survival of cells expressing G466V during therapeutic stress (Fig. 6g).

Based on the role of *BRAF* activity in therapeutic resistance, we predicted that MEKi can function as a sensitizer in cell lines containing hypermorphic variants. Cells with hypermorphic mutations in *BRAF* (RERFLCAD1 and NCI-H2087) treated with AZD6244 and radiation showed a synergistic decrement in clonogenic survival compared with control (0 Gy) cells (Fig. 7a). NCI-H1651 (neutral *BRAF*^mt) and MOR/CPR (*KRAS*^mt) cells showed significantly less response to MEKi alone or in synergy with radiation. NCI-H2405 (hyperactivating *BRAF*^mt) cells, which responded well to MEKi alone, did not show interactive cell death when combined with radiation. These results indicate that treatments that antagonize MEKi can be potent therapeutic

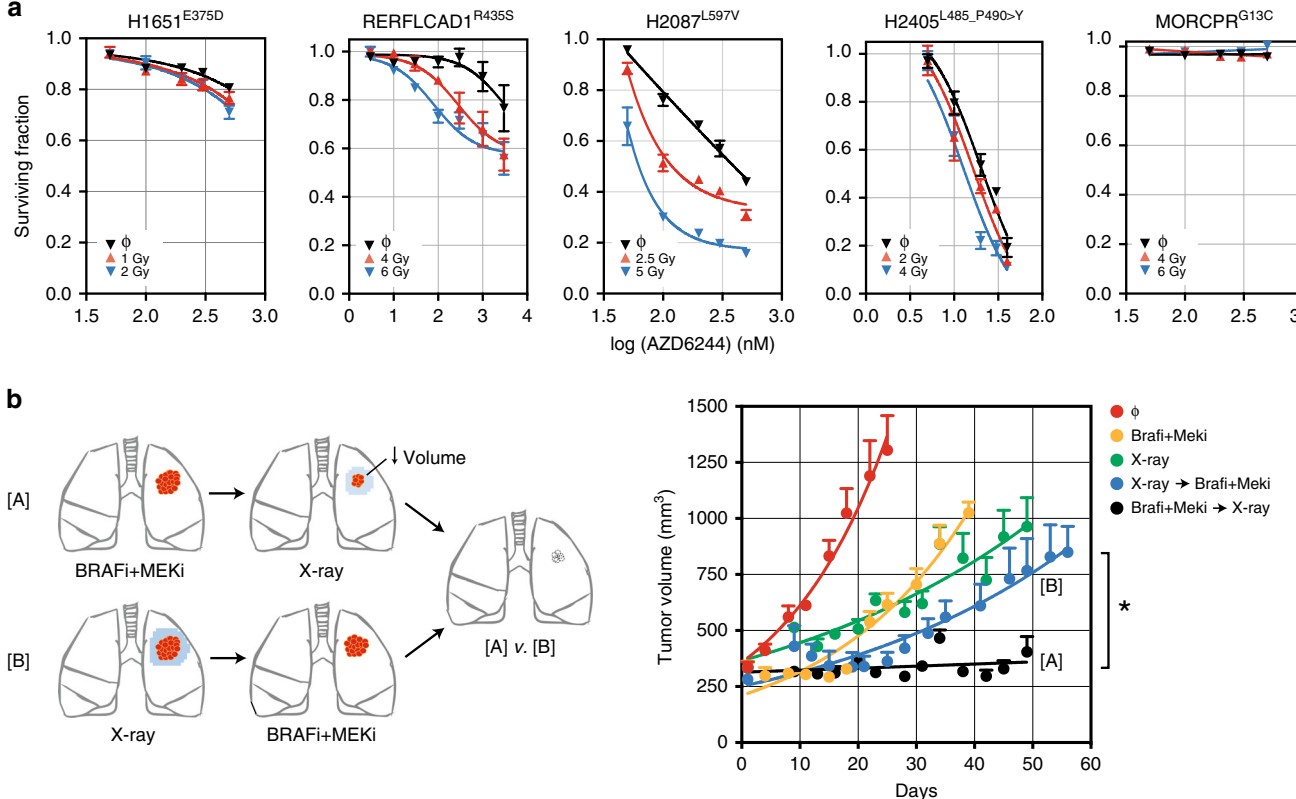

**Fig. 7** Optimal therapeutic strategies for categories of hypermorphic *BRAF* variants in LUAD. **a** Cells were incubated with AZD6244 for 24 h and treated as control (0 Gy) or with radiation. Survival is measured by proliferation assay. Data points represent mean ± s.e.m. **b** Schematic depicting sequential treatment strategies for hyperactivating mutations in *BRAF*. NSG mice bearing LUAD PDX with V600E mutation in the flank were block randomized into one of five treatments arms as shown. Dabrafenib and trametinib were given together for 14 days. Radiation (X-ray) was delivered over three consecutive days to a total dose of 6 Gy. Sequential treatments were given at least 24 h after the completion of the first therapy. Data are expressed as the mean ± s.e. m; *n* = 5 independent animals for each arm. The *P*-value of the χ2-test between [A] and [B] was <0.001

sensitizers in cells containing categories of mutations in *BRAF*, nominating combinatorial treatments as an effective treatment strategy in these tumors.

Since tumors strongly dependent on *BRAF* activity did not demonstrate significant interactive cell death with DNA damage (e.g., NCI-H2405), we posited that sequential rather than concurrent treatment may optimize tumor control, limit toxicity (non-overlapping therapies) and potentially prevent treatment resistance by eradicating minimal residual disease. We tested this hypothesis using a LUAD PDX with a V600E mutations that demonstrated a response to BRAFi/MEKi (Fig. 5b). The PDX was passaged by single-step propagation into 25 mice representing five cohorts that received treatment with either drug alone, radiation alone, drug followed by radiation or radiation followed by drug. Drug followed by radiation was the most effective sequence of therapy that delayed tumor growth (Fig. 7b).

## Discussion

The clinical responses in *BRAF*-driven tumors vary in a manner not fully explained by molecular nosology alone. Our study suggests that intratumoral heterogeneity contributes to this variance and that a more nuanced approach to predict treatment responses is needed. We demonstrate distinct global tumor architecture on the basis of the identity of the mutation and its selective amplification, both of which are influenced by the tumor's tissue of origin. We formalize our observations by translating genomic data into clonal reconstructions and quantitation of subclonal dynamics. Namely, we show that some *BRAF* variants undergo a hard sweep to fixation, resulting in a linear

evolutionary trajectory. We demonstrate that such trajectories are associated with a collapse of genetic diversity and a strong propensity for a dominant subclone, resulting in a greater threat to tumor extinction (i.e., better response to BRAF-pathway inhibition). We provide experimental evidence, in cells, PDX and human tumors, that associate higher *BRAF*-variant multiplicity with less genetic diversity and better response to BRAFi/MEKi. Critically, higher *BRAF*-variant levels appear to predict for prolonged progression free survival in SKCM patients receiving BRAFi or BRAFi/MEKi, lending additional clinical relevance to our explanatory models and preclinical data[45–47].

We also modeled the effect of amplification of distinct *BRAF* variants exposed to targeted therapy. The association between *BRAF*[V600E] fitness and amplification is non-linear and convex (i.e., U-shaped). Previous work has suggested a threshold for *BRAF*[V600E] to regain fitness in the presence of therapy[35]. We show that in addition to this fitness threshold, there is antecedent optimization of allelic fitness that also regulates initial responses to targeted inhibition. Such optimization offers a rationale for the frequent amplification of *BRAF*[V600E] in treatment naïve SKCM tumors. In addition to allelic multiplicity optimization, we note that the extent of *BRAF* amplification required to confer resistance varies in a manner dependent on the level of activation of MAPK pathway. That is, variants with lower levels of pathway activation did not demonstrate resistance to BRAFi/MEKi despite substantial amplification. This suggests that amplification is less likely to be a major mechanism of resistance in some tumors with low-activity *BRAF* variants.

Estimates of the selection, or effect size, of well-known drivers have been previously reported[13,48]. These methods, however, are

dependent on the prevalence of individual variants in a cancer type and, therefore, may not be sufficiently powered to predict selection for rarer drivers. The vast majority of non-V600E mutations in *BRAF* occur at very low frequencies within and across cancer types (Fig. 1a). Despite their rare occurrences, phenotypic profiling of some these variants indicated functional significance as defined by pathway activation and growth (Fig. 1e, f). Therefore, a critical advantage of our framework is the ability to estimate selection based on an individual patient's bulk sequencing data independent of a variant's prevalence in cancer.

Activating BRAF mutants signal as either active monomers (class 1), constitutively active dimers (class 2), or as kinase-dead or -impaired mutants that are RAS-dependent (class 3). Class 1 and 2 mutant signaling has been shown to be effectively blocked by BRAFi, whereas class 3 mutants appear to be insensitive to BRAFi but sensitive to MEKi[24]. Critically, we used BRAFi and/or MEKi in a manner informed by the class of the variant such that downstream Raf-Mek signaling was effectively blocked in an variant-independent manner. This permitted us to examine the effects of distinct classes of variants on tumor response without the confounding effects of the specific molecular interactions between variant types and individual BRAFi.

Using this approach, we demonstrated a difference in treatment response between *BRAF*-driven tumors with variants that confer hyperactivation and selective variant amplification compared to other intermediate- or low-activity mutations. The latter categories retain functional therapeutic relevance in that they confer partial sensitivity to targeted treatments and resistance to DNA damaging therapies. Clinical data are limited in tumors with non-V600 *BRAF* variants and preclinical data have been inconclusive[21–23]. Several ongoing trials seek to evaluate novel strategies in patients with *BRAF*-mutant NSCLC, including patients with distinct classes of non-V600E mutations[49]. Our results suggest that tumors driven by these variants are more likely to respond to combinatorial therapeutic strategies that include DNA damaging agents.

The identification of linear evolution in tumors without selective pressure (i.e., drug treatment) has been elusive to date. The phylogenetic trees of SKCM samples with amplification of *BRAF* show a major dominant clone, with only rare intermediates that are persistent from the previous selective sweeps, consistent with a linear evolutionary process. It remains unclear whether these tumors undergo iterative selective sweeps and, if so, what the underlying molecular basis of this process maybe. It is possible that gradual or punctuated amplification at the *BRAF* locus may contribute to a continuous or interrupted sweep dynamic, respectively. This could give rise to a linear evolutionary phylogenetic pattern and forestall a reversion to a branched or neutral dynamic once a subclone has fixated.

There are limitations to measuring variant subclonal sweep dynamics from bulk sequencing data. First, only sweeps that cause significant changes in the variant frequency distribution can be estimated, leaving the possibility that some *BRAF* variants under selection are not identified using this approach. Second, we estimate selection by investigating the patterns a sweep imprints on a very recent variation that arises during the sweep. Therefore, past events that may have shaped the ancestral diversity of the tumor or the frequency of reversion to branched or neutral evolution are not adequately captured[50]. This is an inherent limitation in the inference of evolutionary history from single time-point samples. Serial tumor samples, either directly from patients or using surrogate experimental models (e.g. PDX), are needed to assess the temporal stability of the evolutionary processes. Third, tumors with uniform rapid population expansion or geographic stratification constraints can influence selection and, in some instances, prevent fixation[51]. This makes it difficult to estimate putative regional sweep dynamics

from bulk sequencing data alone. Lastly, therapy, mainly via cell loss, can alter the neutral mutation rate thereby impacting the accuracy of sweep strength estimates for tumors sampled while on therapy and analysed using these methods. Nonetheless, for tumors without these constraints our models represent an accurate schema in which to interpret driver selection from cancer genomic data.

In summary, we use a quantitative framework to infer the sweep dynamic of individual oncogenic variants and estimate their effects on global tumor architecture. This represents a critical step toward associating the type of cancer evolution in a tumor with the probability of tumor extinction during treatments.

## Methods

**Cell culture**. Cell lines from the Cancer Cell Line Encyclopedia (CCLE) were authenticated per CCLE protocol[52] and grown in recommended media supplemented with 10% fetal bovine serum (ThermoFisher, MA) and 100 U/mL Penicillin, 100 µg/mL of Streptomycin, and 292 µg/mL L-Glutamine (Corning, NY). Adenovirus-12 SV40 hybrid transformed bronchial epithelial cells BEAS2B cells were grown in advanced DMEM-F12 media (ThermoFisher, MA) supplemented with 1% fetal bovine serum and 100 U/mL Penicillin, 100 µg/mL of Streptomycin, 292 µg/mL L-Glutamine and 1% HEPES. All cultures were maintained at 37 °C in a humidified 5% $CO_2$ atmosphere and tested to ensure absence of *Mycoplasma*.

**Antibody and reagents**. Anti-actin 8H10D10 (CST-3700 at 1:4000 dilution), anti-phospho-MEK1/2 41G9 (CST-9154 at 1:1000 dilution), anti-phospho-p44/42 MAPK or Erk1/2 (CST-9101 at 1:3000 dilution), anti-phospho-c-Raf Ser 259 (CST-9421 at 1:1000 dilution), and anti-BRAF D9T6S (CST9421 at 1:1000 dilution) were from Cell Signaling Technology (Beverly, MA). Anti-vinculin Ab-1 VLN01 (MS-1209-PO at 1:4000 dilution) was from Thermo-Fisher Scientific (Waltham, MA). AZD6244, trametinib and dabrafenib were from Selleckchem (Houston, TX). Doxycycline hyclate and puromycin were from Sigma (St. Louis, MO).

**BRAF alignment and three-dimensional structure mapping**. Clustal Omega (version 1.2.4) was used for multiple BRAF protein sequence alignment. The three-dimensional model of BRAF in complex with dabrafenib was used to map genetic variants to the protein structure[53].

**Variant generation in lentiviral vectors**. We performed high-throughput mutagenesis in three steps: PCR, in vitro recombination and transformation (primer sequences are in Supplementary Data 2). Briefly, the BRAF ORF was PCR amplified by using primers that contain incorporated mutated sequence. Fragments were the transferred directly to the destination vector (pLX or pCW57.1) by LR reaction (Invitrogen) and the constructs were transformed into competent cells. The discontinuity at the mutation site was repaired by endogenous bacterial repair mechanism. After virus infection (multiplicity > 1), BEAS-2B cells were selected and maintained in the presence of 1 µg/mL puromycin.

**BRAF gene expression signature**. For each gene from a dataset of total mRNA, we normalized expression values to standard deviations from the median across samples. The most variable genes were identified by calculating the median absolute deviation. The gene list was pruned to 153 on the basis of several gene signature quality metrics including signature gene variability, compactness (as measured by gene autocorrelation) and by the proportion of variance attributable to the first principal component[54]. Since the signature had both "up" and "down" genes, we calculated, within each sample, the size of the difference of the "up" and "down" genes relative to the variation in each sample to determine the BRAF score. A score that estimates ERK activity on the basis of the BIOCARTA_ERK_PATHWAY (http://software.broadinstitute.org/gsea/msigdb/index.jsp) gene set was also computed.

**Mouse studies**. NSG mice were bred in the Cleveland Clinic Biological Resources Unit facility. All mouse studies were conducted under a protocol approved by the Cleveland Clinic Institutional Animal Care and Use Committee. BEAS-2B infected with a lentiviral vector (pLX302, pLX306, or pLX307) expressing backbone or BRAF were injected into the flank of NSG mice and monitored for growth. Tumor volume was calculated using the formula: (length × width²)/2[55]. PDX were developed using a sample collection protocol approved by the Institutional Review Board at the Cleveland Clinic. Biological material was obtained from patients who provided written informed consent. The sample collection protocol was approved by the Institutional Review Board at the Cleveland Clinic and complied with all relevant ethical regulations. Tumors were mechanically-processed into sub-millimeter pieces in antibiotic-containing RPMI medium, combined with Matrigel and implanted into the flank of a 6–8-week-old female NSG mice using a syringe with a 20 G needle. Tumors were harvested and stored for biological assays on reaching a size of >1000 mm³. Mice were randomized into treatment arms when tumors reached ~200–300 mm³ in volume. Drugs were formulated according to the

manufacturer's specifications. To establish if intergroup differences were significant, we used regression with random effect and autoregressive errors (RE/AR)[56]. A likelihood ratio test was used to compare the null model (assumes same slope in each group) to the alternative model (assumes different slopes in each group) to assess differences between treatment groups. A P-value of <0.05 associated with the χ2-test was considered to be statistically significant.

**Cancer cell fraction**. Gene-level copy-number and mutational data were combined with estimates of the sample purity to infer the cancer cell fraction, or the proportion of cancer cells with the single-nucleotide variant ($CCF_{SNV}$) as follows:

$$CCF_{SNV} = \frac{VAF * (2 + (ploidy_{CNV} - 2) * CCF_{CNV})}{purity} \quad (4)$$

where VAF is the fraction of sequencing reads overlapping a genomic coordinate that support the non-reference allele. Ploidy is the copy number of the locus affected by an overlapping copy-number variant (CNV). $CCF_{CNV}$ is the fraction of cells affected by the CNV and purity represents the fraction of tumor cells in the sequenced sample. Ploidy was estimated from the log2 ratio segment means from the Affymetrix SNP 6.0 array data. Purity was estimated using ESTIMATE[57]. CCF correction heuristics were applied (see Supplementary Software 1)[58]. Each SNV was classified as clonal if the $CCF_{SNV}$ exceeded 0.95.

**Subclone reconstruction**. Subclone reconstruction was performed using *phyloWGS*[27]. Briefly, up to 2500 sampled trees were calculated for each tumor using Markov chain Monte Carlo settings. Trees were ranked using a normalized *log* likelihood to determine the solutions that best describe the input. The linearity and branching indices are summary values of the sampled trees and represent the extent that the proportion of mutations are in linear or branched relations, respectively. Sequencing error rates were assumed to be uniform across the genome.

**Diversity index**. We divided the CCF histogram into 10 equal bins. We calculated the information entropy (or Shannon index)[59], which can be used to estimate diversity in a biological sample, as follows:

$$H = -\sum_{i=1}^{10} p_i \ln p_i \quad (5)$$

where $p_i$ is the proportion of mutations in bin $i$.

**Cell survival measurements**. *High-throughput proliferation assay*: Cells were plated using a Multidrop Combi liquid handler (Thermo Fisher) in at least quadruplicates for each time-point at three cell densities (range 25–225 cells/well) in a white 384-well plate (Corning, NY). Plates were irradiated and at 7–12 days post-irradiation, media was aspirated and 40 μL of CellTiter-Glo® reagent (50% solution in PBS) (Promega, WI) was added to each well. Relative luminescence units were measured using an Envision multilabel plate reader (Perkin Elmer) with a measurement time of 0.1 s. Luminescence signal is proportional to the amount of ATP present. For chemical radiosensitization measurements, drug was added 24 h prior to irradiation. The luminescence signal was plotted as a function of cell density and a cell density within the linear range for luminescence (or growth) was selected to generate integral survival for each cell line[43].

*Clonogenic survival*: Cells were plated at appropriate dilutions, irradiated, and incubated for 7–21 days for colony formation. For chemical radiosensitization measurements, drug was added 24 h prior to irradiation. Colonies were fixed in a solution of acetic acid and methanol 1:3 (v/v) and stained with 0.5% (w/v) crystal violet[60]. as previously described[60]. A colony was defined to consist of 50 cells or greater. Colonies were counted digitally using ImageJ software as described[61]. Integration of survival as a function of dose, or area under the curve, was calculated using Prism, GraphPad Software (La Jolla, CA).

**Western blot analysis**. Whole-cell lysates were prepared using M-PER lysis buffer and clarified by centrifugation. Proteins were separated by SDS–PAGE and transferred onto 0.45 μM nitrocellulose membranes (Maine Manufacturing; Sanford, ME). After primary antibody incubation for 1–2 h at room temperature, washings, and incubation with secondary antibodies, blots were developed with a chemiluminescence system (Amersham/GE Healthcare). Full images of cropped blots are shown in Supplementary Fig. 13.

**Droplet digital PCR**. Dye-labeled sequence-specific oligonucleotide (TaqMan) were used in singleplex assays. The Bio-Rad QX200™ PCR System was used for both droplet generation and variant detection. Each droplet was assigned a binary readout of "positive" or "negative," which is used to determine the existence of the target DNA. The fractional abundance was calculated using the Quantasoft™ software.

**Genetic data**. *BRAF-variant profiling*: BRAF-variant frequencies were calculated using genome-wide screen data from v81 of COSMIC, the Catalogue of Somatic Mutations in Cancer (http://cancer.sanger.ac.uk). *BRAF*-variant expressing cells

(BEAS-2B) were profiled for gene expression using RNA sequencing (RNAseq). Total RNA was converted to mRNA libraries using the Illumina mRNA TruSeq kit following the manufacturer's directions. Libraries were sequenced on the Illumina HiSeq 2500. RNA reads were aligned to the hg19 genome assembly using STAR[62]. Read counts were normalized within-sample and *log* transformed. Genetic data profiled by TCGA, including exome and transcriptome sequencing and copy-number estimates, were obtained from the Firebrowse (http://firebrowse.org/). Exome capture was performed using paired-end sequencing on the Illumina HiSeq platform. Transcriptome analysis was performed using RNAseq and gene expression was quantified using RSEM and normalized within-sample to a fixed upper quartile. The Genomic Identification of Significant Targets in Cancer (GISTIC) algorithm ENREF 66[63] was used to identify focal regions of copy-number alterations in individual samples. A gene-level copy-number was also generated, defined as the maximum absolute segmented value between the gene's genomic coordinates, and calculated for all genes using the hg19 coordinates provided by the refFlat and wgRna databases from UCSC Genome Browser (http://hgdownload.cse.ucsc.edu/goldenPath/hg18/database/). Cancer cell lines were profiled at the genomic level and processed as described in detail[52]. The processed data is available for download at http://www.broadinstitute.org/ccle.

**Irradiation**. Cells were treated with γ-radiation delivered at 0.90 Gy/min with a [137]Cs source using a GammaCell 40 Exactor (Best Theratronics; Ontario, Canada). Mice were irradiated using a 320 kVp orthovoltage machine (XRAD-320, Precision X-ray) at a dose/rate of 3.0 Gy/min and a source to skin distance of 50 cm, through a 1 mm copper filter for standard fraction treatment. Mice were anesthetized and a lead shield with a circular opening was placed over mice receiving flank tumor-directed radiation. For quality assurance, thermoluminescent dosimeters were used to verify correct dose administration.

**Information-based association metric**. The association between genomic alterations (e.g., *BRAF* mutations) and the radiation response profile was determined using the Information Coefficient (IC)[43,64,65]. This quantity is obtained by estimating the differential mutual information between the radiation response profile in cells with and without *BRAF* variants. The nominal P-values for the information-based association metric between the genetic parameters (alterations) and radiation response values were estimated using an empirical permutation test.

**Reporting summary**. Further information on research design is available in the Nature Research Reporting Summary linked to this article.

## Data availability

The RNA sequencing data have been deposited in the GEO database under the accession code GSE133151. DNA sequencing and copy-number data are available from the Genomic Data Commons portal (https://gdc-portal.nci.nih.gov). Drug sensitivity data are available from the Cancer Target Discovery and Development (CTD²) initiative (https://ocg.cancer.gov/programs/ctd2). Genomic data pertaining to cancer cell lines were downloaded from the Cancer Cell Line Encyclopedia (http://www.broadinstitute.org/ccle). All of the other data supporting the findings of this study can be found in GEO (www.ncbi.nlm.nih.gov/geo/) using accession code: GSE133151. Other datasets analyzed during the current study are available within the article, its supplementary files, or from the corresponding author upon reasonable request.

## Code availability

All custom or modified code can be accessed in Supplementary Software. There are no restrictions to access.

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

## Acknowledgements

M.E.A. was supported by NIH KL2 TR002547, NIH R37 CA222294, the American Lung Association and VeloSano. E.K.C. was supported by SNUH Research Fund 0420160010.

## Author contributions

P.G. conducted and analyzed the experimental work. E.I.S and S.D. assisted with the computational work. E.K.C. and G.K. provided experimental support. N.A.P., D.P.R., and F.A. assisted with analysis and interpretation. S.C.M., U.A., S.R., S.S., and T.R.G. assisted with data acquisition. C.D.P. and D.J.A. assisted with interpretation and edited the manuscript. M.E.A. conceived, designed, analyzed, interpreted, and supervised the experimental and computational work. M.E.A. wrote the manuscript.

## Competing interests

M.E.A. receives grant support, travel support, and honoraria from Bayer AG and receives grant support from Siemens Medical Solutions, USA in subject matter or material not directly related to this work. The other authors disclose no potential conflicts of interest.

**Additional information**

**Supplementary information** is avaliable for this paper at https://doi.org/10.1038/s41467-019-13161-x.

