## [Peer Review File · Nature Communications]

Reviewers' comments:

Reviewer #1 (Remarks to the Author):

This manuscript digs deep into the evolutionary dynamics of BRAF mutations, showing the very interesting result that strongly activating BRAF mutations demonstrate hard sweep dynamics whereas mutations with less pronounced activation of the BRAF signaling pathway confer soft sweeps or subclonal. The manuscript marshals a dauntingly impressive array of experiments, data, and analyses to elaborate on this interesting finding. In my evaluation, it is one of the best examples that I've read in which evolutionary thinking is (unusually!) appropriately applied to the problems of cancer progression. It has a few moderate weaknesses—mostly regarding clarity and precision of communication in specific instances—that should be remedied.

(N.B. please don't interpret any comments below on wording as snide; I'm just trying in some cases to communicate wording issues efficiently)

LINE

42–43: "We provide the means to predict the strength of "driver" selection..." This statement seems too broad; I'm not even sure what part of the manuscript it is describing. What are these "means"? A heuristic? Equations? An experimental protocol? An algorithm? Do you mean to stake this claim for all "drivers" or just some subset? And is "predict" the best word here—if so, in what sense?

50--61: Nice introduction

62–68: Needs a lot of work:

62–63: I'm not sure what is being claimed here, really—and I know it's just citing other work—thus it is hard for me to coherently dispute it, but I think it makes no sense to claim that one can "genotype" individual cells "using population frequencies". This sentence needs to be more granularly written to cogently explain the claim that these references support.

63–64: "these frequencies are represented by the distribution of variant-allele fractions": this phrase seems either abysmally trivial or more complicated than stated.

64–66: Precisely speaking, "Inferences" can't "identify". More importantly (and related), there are a lot of caveats (underlying assumptions) to a one-on-one mapping of relative abundance to magnitude of selection, many of them not generally tenable. I think that's what's being hinted at by the "inferences" phrasing, but please be more explicit. Consider consulting or referencing *1 regarding the roles of tumor subclones.

66–67: calling the _roles of resident molecular drivers_ "variables" (that's what the e.g. says is meant by variables here) is confusing rather than helpful. Don't you just mean "quantifying" the magnitude of a subclone's fitness?

67–68: "the impact of the strength of subclone selection on the tumors' global genetic composition remain[s] unknown": whether this statement is true or not depends on whether you're dedicated to quantifying selection on _specific_ subclones in tumors or whether you're interested in quantifying the average effect of a mutation's impact on selection of the subclone it engenders (which has been done; *2). The introduction should clarify here.

69–71: Nice end

78–80: "unknown significance": a word is needed here to delineate what kind of significance you

mean (mechanistic?) because their significance has been characterized in a variety of ways *3–8, but I think you just mean here that it is unknown whether they are hypermorphic or not?

84–85: “It is not clear whether mutations in these putative BRAF-driven tumors ... alter the tumor genetic composition”: If they are selected, then they are altering the tumor genetic composition, and their average selective effect on the subclones they engender is known *2.

95–96: I understand these statements are in fact reconcilable, but “As expected, ...” here strikingly contrasts with the “approximately half...” statement on lines 80–81. Perhaps you can rewrite these lines in a way that is less jarring to the reader.

97: “multiple lineages” here is confusing, because of the terrible methods-after results structure imposed by many journals. What these lineages you are discussing are needs to be clarified somehow at this point in the manuscript. Same point on line 104 with “lineage preferences”.

109: Did you really procedurally randomize, or was it haphazard?

115: were->was, but there’s still something wrong; a signature doesn’t calculate.

119*: American Statistical Association convention is capital italic P for P value as it is a random variable (throughout); capital italic Q would follow for Q value.

123–126: Are these tumor formation times consistent with previously calculated average selective effects 2?

125: “inversely proportional” is clearer.

136–139: This part is basic population genetics 9. Relevant theory should be referenced. Simulation is not necessary.

142: gene-level

152: For clarity, please follow “This” by a noun or noun phrase. More importantly, although “suggested” is used here, it should be more clearly stated that there are multiple other plausible interpretations.

163–184: Very interesting.

186–200: Very interesting results—and I do believe them! But please consider whether usage of “determine” and “phylogenetic” are best here and in lines 220 and 251. I’m not sure, but they strike me as potentially over-sure and fairly unusual, respectively.

192 “Fig. 3f describes...” Perhaps better to parenthetically reference the figure panel at the end of a more informative sentence.

214: lead->led

232–233: I understand the convenience of doing so and think it’s alright to make the constant N assumption here even though it’s very unlikely, but think some mention should be made that it’s not likely true.

233: defined fixation \hat{t} to be when the fit variant reached \hat{t} , but please clarify why fixation was defined as reaching $1 / 2N_s$. Why not $(2N-1) / 2N$? Why should the definition of fixation be dependent on s ?

238, 240, 255: low-frequency

256: ref should be after date, not type, so that it isn't misleading.

254–256: I agree that it suggests these low-frequency variant differences aren't a consequence of neutral mutation (that is what is being said here?), but wouldn't it be sensible just to attribute the low-frequency variant differences more directly to the high magnitude of selection of somatic BRAF V600E? High mutation load in skin, but low diversity in tumor.

288–303: Very interesting.

311: more info needed on this information-based similarity index, so the reader can better understand what Fig. 6a means.

—DISCUSSION is excellent. No comments.

452–457: This analysis is OK, but seems very simplistic, statistically speaking. Perhaps some more thought could be put in to a more powerful approach or the signature quality metrics could be explained more granularly so as to strengthen the reader's confidence in the result. Also, ref to 47 should have author et al. as part of sentence.

463–464: Why this formula for tumor volume? Seems very arbitrary. Citation?

475: no hyphen between P and value. "of < 0.05" should follow P value. Associated _to_?

487: "ESTIMATE"—citation?

497–498: Each..."species" is a strange analogy. Probably best to delete this sentence. It isn't needed. Shannon index was not invented for ecology or evolutionary biology.

562–567: is it an "association score" or a Coefficient? There isn't enough information here to interpret what IC is. Please summarize what it is here. Also: "information-based".

714: is "graded" the best word here?

716: "coordinate" is unclear, and is it the majority or the vast majority?

736: needs more explanation of the "trees": can you be more specific how to "read"/interpret them? There are lots of kinds of trees; what are these? How does one read the panel nodes, colors, sizes, lines?

755: t in t-test should be italic throughout

757: locus-specific...single-nucleotide variant

789: Perhaps there is a better way than an empirical permutation test—one that specifies a null hypothesis. If not, clarify how permutation was performed and null that permuted data represents.

Figure 1a: y-axis is unclear.

Very interesting manuscript! —Jeffrey Townsend

* References cited

1. Ryu, D., Joung, J.-G., Kim, N. K. D., Kim, K.-T. & Park, W.-Y. Deciphering intratumor heterogeneity using cancer genome analysis. *Human Genetics* 135, 635–642 (2016).
2. Cannataro, V. L., Gaffney, S. G. & Townsend, J. P. Effect Sizes of Somatic Mutations in Cancer. *J. Natl. Cancer Inst.* 110, 1171–1177 (2018).
3. Pupo, G. M. et al. Clinical significance of intronic variants in BRAF inhibitor resistant melanomas with altered transcript splicing. *Biomark Res* 5, 17 (2017).
4. Xu, Y. et al. Low frequency of BRAF and KRAS mutations in Chinese patients with low-grade serous carcinoma of the ovary. *Diagnostic Pathology* 12, (2017).
5. Dahlman, K. B. et al. BRAF(L597) mutations in melanoma are associated with sensitivity to MEK inhibitors. *Cancer Discov.* 2, 791–797 (2012).
6. Sheen, Y.-S. et al. Prevalence of BRAF and NRAS mutations in cutaneous melanoma patients in Taiwan. *Journal of the Formosan Medical Association* 115, 121–127 (2016).
7. Berger, A. H. et al. High-throughput Phenotyping of Lung Cancer Somatic Mutations. *Cancer Cell* 32, 884 (2017).
8. Cardarella, S. et al. Clinical, Pathologic, and Biologic Features Associated with BRAF Mutations in Non-Small Cell Lung Cancer. *Clinical Cancer Research* 19, 4532–4540 (2013).
9. Hartl, D. L. & Clark, A. G. *Principles of Population Genetics*. (Sinauer Associates Incorporated, 2007).

Reviewer #2 (Remarks to the Author):

This is a well articulated and coherent paper that makes a coherent case that different BRAF mutation drive different evolutionary trajectories in tumours. They provide well developed and clearly presented experimental data that the BRAF mutations are causal, as well as appropriate analysis of publicly available patient data.

All the experiments were well explained in both motivation and significance of the results, and the paper provides a good synthesis of the overall results.

The clinical implications are explored with an appropriate preclinical model and support the potential significance of the results.

Minor comments

The TCGA data analysis relies to some degree on similar mutation rates across tumour types, and makes a clock like assumption of mutation that is potentially violated by therapy. It would be good to make mention of this, perhaps in the discussion.

p9 line 189-190 the authors discuss strand. I assume this is meant to be allele? If they wish to avoid the word allele strand is not a good alternative. Copy or template would be preferable, but in my opinion allele or somatic allele would be preferable

It would be preferable if the RNAseq data could be deposited in an appropriate public repository, rather than being available solely by request. If it is to be available only by request it would be ideal if this could be handled by an institutional data access committee.

Reviewer #3 (Remarks to the Author):

The paper by Gopal et al is a very well written description of the role that different BRAF mutations play in terms of their contribution to activation of downstream pathways, tumour growth and clonal evolution. Overall the paper is interesting and the data is beautifully presented. One could

argue, however, that it doesn't really tell us something we didn't already know. i.e. that V600E is a potent oncogenic mutation and most other mutations are weaker. Many of the mutations shown have been tested in biochemical/functional assays so we knew this already (1). We also knew that different mutations are associated with different clonal behaviour/subclonality (2), although this paper illustrates this more systematically. Likewise, we know that BRAF mutant tumours (melanomas) have a different genomic landscape compared to tumours without BRAF mutations – this in part is because they can be different melanoma subtypes (the authors appear to group all TCGA subtypes together in their analysis (cutaneous/acral/mucosal) (3). We also already knew that the BRAF locus is amplified in tumours (melanomas) that are BRAF mutant (4) and it is no surprise that amplification is associated with expression differences. The observation that “higher” BRAF variants predict longer progression free survival is also well established (4). Likewise, BRAF mutations occur early (often in nevi) so must, in most cases, proceed amplification of the BRAF locus (5). The authors should be commended for a very detailed job of joining all of these threads together, and writing a paper that is glorious to read, but I don't see the novelty in terms of biological insights.

1. PMID: 30096302
2. PMID: 29748584
3. PMID: 26091043
4. PMID: 29625053
5. PMID: 23690527

Reviewer #4 (Remarks to the Author):

Gopal and colleagues studied the selection and propagation of BRAF mutation in cancer. They found that V600 mutations hyperactive the ERK signaling pathway and are represented at a higher cancer cell fraction in tumors, as opposed to non-V600 mutations, which are have sub clonal patterns based on CCFs. They also go on to identify evolutionary lineages modeling the evolution of these mutations and correlate these patterns with response to therapy.

While the study is interesting and makes some potentially interesting observations there are several limitations.

1. The authors generalize their findings across all BRAF mts yet they only prospectively test V600E or G466V mts. More representatives from each type of BRAF mutations should be tested.
2. The authors use transcriptional output to score the activity of BRAF mutants. The signature score seems to have been derived from expressing BRAF mts in an immortalized lung cell line. Wouldn't this signature be different if the mutants were expressed in another lineage? Do the mutations rank in a similar fashion if the authors were to use published ERK output signatures to order them?
3. The authors should compare the clonal trajectory of the V600E mt form lung, colon and melanomas and determine if these are always clonal regardless of lineage. The way that the text is written makes it difficult to conclude if the clonal pattern of this mutation is due to its biochemical properties or due to lineage-specific factors.
4. The authors generated isogenic cell lines of 35 variants and correlated high output scores with faster tumor formation. Did all variants have the same protein expression level?
5. The authors show that BRAF V600E amp increases tumor growth rate in the absence of therapy. This is not consistent with data showing that BRAF amp is at a fitness disadvantage in absence of

treatment. This needs to be addressed.

6. What does a 'moderately hypermorphic variant' mean? G466V has been described on several occasions to be a low activity mutant. This mutation transforms by dimerizing with WT RAFs. Why are the authors surprised that induced expression of G466V doesn't activate ERK in the same dose dependent manner as V600E? Wouldn't the effect of G466V (regardless of how much of this is expressed) be limited by the level of endogenous A/B/CRAF expression (since the later are required to dimerize and provide the necessary kinase activity to stimulate the ERK pathway)? This experiment sheds little light into the effects of amplification on fitness. The authors should compare the induced expression of V600E to the induced expression of other high activity mutants.

7. They show that BRAF amp is sensitive to RAF/MEK inhibitor combination. What is the level of amplification at which this effect was observed? It seems that at higher levels of dox induced BRAF expression the tumors grow while on this treatment (Fig 5d). What is the rationale for testing the RAF/MEK inhibitor combination in non V600E mutant settings? These should not respond to the RAF inhibitor. Did the authors use a dimer inhibitor?

Minor-

1. The experimental work should be described better, particularly that describing the effect of amplification, RAF/MEK combination and DNA damage inducing agents. The authors spend a lot of time describing computational analysis, and this comes at the price of an appropriate description of the experimental work.

2. Some of the ideas in the manuscript could be conveyed in a simpler and more direct manner to the reader.

We thank the reviewers for taking the time to carefully read the manuscript and provide these thoughtful critiques. We address the reviewers' comments and suggestions in detail below.

Reviewers' comments:

Reviewer #1 (Remarks to the Author):

This manuscript digs deep into the evolutionary dynamics of BRAF mutations, showing the very interesting result that strongly activating BRAF mutations demonstrate hard sweep dynamics whereas mutations with less pronounced activation of the BRAF signaling pathway confer soft sweeps or subclonal. The manuscript marshals a dauntingly impressive array of experiments, data, and analyses to elaborate on this interesting finding. In my evaluation, it is one of the best examples that I've read in which evolutionary thinking is (unusually!) appropriately applied to the problems of cancer progression. It has a few moderate weaknesses—mostly regarding clarity and precision of communication in specific instances—that should be remedied.

We thank the reviewer for stating that the manuscript "...is one of the best examples that I've read in which evolutionary thinking is (unusually!) appropriately applied to the problems of cancer progression" and that the manuscript "...marshals a dauntingly impressive array of experiments, data, and analyses to elaborate on this [clonal selection] interesting finding".

(N.B. please don't interpret any comments below on wording as snide; I'm just trying in some cases to communicate wording issues efficiently)

We greatly appreciate the reviewer's comments and suggestions; they have significantly improved the clarity of the manuscript.

LINE 42–43: "We provide the means to predict the strength of "driver" selection..." This statement seems too broad; I'm not even sure what part of the manuscript it is describing. What are these "means"? A heuristic? Equations? An experimental protocol? An algorithm? Do you mean to stake this claim for all "drivers" or just some subset? And is "predict" the best word here—if so, in what sense?

The "means" here refers to using driver genotype and clonal reconstructions. We agree that "predict" is imprecise. We have modified the sentence as follows:

"We use clonal reconstructions to estimate the strength of "driver" selection in individual tumors using available sequencing data."

50–61: Nice introduction

Thank you.

62–68: Needs a lot of work:

62–63: I'm not sure what is being claimed here, really—and I know it's just citing other work—thus it is hard for me to coherently dispute it, but I think it makes no sense to claim that one can "genotype" individual cells "using population frequencies". This sentence needs to be more granularly written to cogently explain the claim that these references support.

63–64: "these frequencies are represented by the distribution of variant-allele fractions": this phrase seems either abysmally trivial or more complicated than stated.

Changed to: “In some cases, it is possible to infer the genotype of these groups of cells using the population frequencies of mutations that distinguish them.”

*64–66: Precisely speaking, “Inferences” can’t “identify”. More importantly (and related), there are a lot of caveats (underlying assumptions) to a one-on-one mapping of relative abundance to magnitude of selection, many of them not generally tenable. I think that’s what’s being hinted at by the “inferences” phrasing, but please be more explicit. Consider consulting or referencing *1 regarding the roles of tumor subclones.*

We propose here that the average frequency of a subclonal cluster of mutations encodes the size of the subclone, which in turn allow measurement of the clone’s selective advantage. We concede that the relationship is not one-to-one. We have changed the sentence to the following: “The relative size of a subclone, and relatedly its selection, could be estimated from the average of its VAF cluster.¹” We have also consulted and referenced *1.

66–67: calling the _roles of resident molecular drivers_ “variables” (that’s what the e.g. says is meant by variables here) is confusing rather than helpful. Don’t you just mean “quantifying” the magnitude of a subclone’s fitness?

Yes, many thanks; that is precisely what we mean. This has been changed.

*67–68: “the impact of the strength of subclone selection on the tumors’ global genetic composition remain[s] unknown”: whether this statement is true or not depends on whether you’re dedicated to quantifying selection on _specific_ subclones in tumors or whether you’re interested in quantifying the average effect of a mutation’s impact on selection of the subclone it engenders (which has been done; *2). The introduction should clarify here.*

The work by Cannataro *et al* is significantly related to aspects of this manuscript. It was an (unintentional) oversight to not include this highly relevant work in our reference list. Here, we meant the former. Changed to: “However, quantifying the magnitude of a subclone’s fitness (*e.g.* the role of resident molecular drivers) and the impact of the strength of a subclone’s selection on the global genetic composition of a tumor remains a challenge. In cases in which the genetic drivers that regulate selection in an individual tumor can be targeted with a therapy, delineating the relationship between evolutionary processes and the probability of tumor extinction can prove decisive in guiding treatment strategies.”

We have also modified the **DISCUSSION** as follows: “Estimates of the selection, or effect sizes, of well-known drivers have been previously reported.^{2,3} These methods, however, are dependent on the prevalence of individual variants in a cancer type and therefore, may not be sufficiently powered to predict selection for rarer drivers. The vast majority of non-V600E mutations in BRAF occur at very low frequencies within and across cancer types (Fig. 1A). Despite their rare occurrences, phenotypic profiling of some these variants indicated functional significance as defined by pathway activation and growth (Fig. 1E-1F). Therefore, a critical advantage of our framework is the ability to estimate selection based on an individual patient’s bulk sequencing data independent of a variant’s prevalence in cancer.”

This and similar work are now cited in the manuscript.

69–71: Nice end

Thank you.

78–80: “unknown significance”: a word is needed here to delineate what kind of significance you mean

*(mechanistic?) because their significance has been characterized in a variety of ways *3–8, but I think you just mean here that it is unknown whether they are hypermorphic or not?*

We meant clinical. We have added clinical before “significance.”

*84–85: “It is not clear whether mutations in these putative BRAF-driven tumors ... alter the tumor genetic composition”: If they are selected, then they are altering the tumor genetic composition, and their average selective effect on the subclones they engender is known *2.*

Agreed. We have removed “...or alter the tumor genetic composition.”

95–96: I understand these statements are in fact reconcilable, but “As expected, ...” here strikingly contrasts with the “approximately half...” statement on lines 80–81. Perhaps you can rewrite these lines in a way that is less jarring to the reader.

We have removed “As expected.”

97: “multiple lineages” here is confusing, because of the terrible methods-after results structure imposed by many journals. What these lineages you are discussing _are_ needs to be clarified somehow at this point in the manuscript. Same point on line 104 with “lineage preferences”.

We agree with the reviewer that “lineage” needs to be clarified, especially since we also use lineage in the context of clonal expansion in a tumor undergoing a selective sweep. We have changed “multiple lineages” to “cancer types” and reserved the singular use of “lineage” in the manuscript to describe subclone lineage (line 225).

109: Did you really procedurally randomize, or was it haphazard?

Yes, we used Excel’s RAND function to randomly rearrange the selected list of variants.

115: were->was, but there’s still something wrong; a signature doesn’t calculate.

The signature now “...estimates BRAF activity.”

119: American Statistical Association convention is capital italic P for P value as it is a random variable (throughout); capital italic Q would follow for Q value.*

Thank you. These changes have been made.

123–126: Are these tumor formation times consistent with previously calculated average selective effects 2?

The short answer is yes—generally. The long answer is as follows: There is support in our experimental data for the cancer effect size predicted by the selection intensity. We say generally because there is limited overlap (V600E, G466V, G469A) in variants assessed in the two datasets: our results and data from **Supplementary Table 3** in Cannataro, VL, *et al.*² Despite this, an assessment of the overlap that exists revealed a rank order trend between the BRAF score and selection intensity and, by extension, the CCF and selection intensity. For example, the selection intensity in BRAF V600E mutant samples across the three cancer types are as follows: SKCM>>COAD>>LUAD. This is precisely in line with our predicted cellular multiplicity and clonal evolution predictions in our analyses. Additionally, the selection intensity rank order in LUAD is as follows: V600E >> G469A>G466V. This is also the order of BRAF score and *in vivo* tumor growth in our bronchial epithelial cell line (BEAS2B).

125: “inversely proportional” is clearer.

We agree. Thank you.

136–139: This part is basic population genetics 9. Relevant theory should be referenced. Simulation is not necessary.

Relevant theory is now referenced. We agree the simulation is not necessary, but considering the diverse readership of *Nature Communications*, we thought it may be helpful for non-specialists to visualize the theory.

142: gene-level

Corrected.

152: For clarity, please follow “This” by a noun or noun phrase. More importantly, although “suggested” is used here, it should be more clearly stated that there are multiple other plausible interpretations.

We agree. Corrected.

163–184: Very interesting.

Thank you.

*186–200: Very interesting results—and I do believe them! But please consider whether usage of “determine” and “phylogenetic” are best here and in lines 220 and 251. I’m not sure, but they strike me as potentially over-
sure and fairly unusual, respectively.*

Agreed. We have changed “determine” to “suggest” and deleted “phylogenetic”.

*192 “Fig. 3f describes...” Perhaps better to parenthetically reference the figure panel at the end of a more
informative sentence.*

Agreed. Changed to: “The orders between gene copy gain and mutation is predicted to differentially impact the VAF at an individual locus (Fig. 3f).”

214: lead->led

Changed.

232–233: I understand the convenience of doing so and think it’s alright to make the constant N assumption here even though it’s very unlikely, but think some mention should be made that it’s not likely true.

Changed to: “Although the model assumes a constant N , most tumors have a growth fraction that can be, in part, balanced by cell loss. Population size changes can make it difficult to distinguish selection from demographic processes, but only in cases in which there is weak selection.⁴”

233: defined fixation ^to be when the fit variant reached^ , but please clarify why _fixation_ was defined as reaching $1 / 2Ns$. Why not $(2N-1) / 2N$? Why should the definition of fixation be dependent on s ?

Thank you for your astute identification of this error. The $n \approx (2Ns)^{-1}$ is the population frequency in which a fit variant is established and not when it achieves fixation. We have re-worked this section as follows: “First, we modeled the frequency trajectory of a new adaptive mutation using logistic growth as follows:

$$n(t) = \frac{e^{st}}{e^{st} + 2Ns} \quad [5]$$

where s is the fitness, t is the time measured in units of generations, N is a population of constant size, and $n \approx (2Ns)^{-1}$ is the population frequency in which a fit variant is established. Although the model assumes a constant N , most tumors have a growth fraction that can be, in part, balanced by cell loss. Population size changes can make it difficult to distinguish selection from demographic processes, but only in cases in which there is weak selection.⁴ Importantly, once a variant is established, it escapes stochastic loss and can be modeled by logistic growth.⁵ To model the frequency trajectory of the i th passenger mutation we used:

$$n_i(t) = e^{-\mu t} \left(\frac{\mu}{is}\right)^{1-\mu/s} \quad [6]$$

where $i \geq 1$, μ is the rate at which neutral mutations occur on the sweeping clone and s is equal to the fitness of the adaptive subclone (Fig. 4b).”

238, 240, 255: low-frequency

Changed.

256: ref should be after date, not type, so that it isn't misleading.

Changed.

254–256: I agree that it suggests these low-frequency variant differences aren't a consequence of neutral mutation (that is what is being said here?), but wouldn't it be sensible just to attribute the low-frequency variant differences more directly to the high magnitude of selection of somatic BRAF V600E? High mutation load in skin, but low diversity in tumor.

We attribute the low-frequency variant differences to the high magnitude of selection of somatic *BRAF* V600E. In this paragraph, we entertain several alternative hypotheses that could explain the low-frequency variant differences independent of selection. Low neutral mutation rate is one alternative hypothesis that we reject due to the high mutational burden in SKCM.

288–303: Very interesting.

Thank you.

311: more info needed on this information-based similarity index, so the reader can better understand what Fig. 6a means.

Changed to: “Genomic correlates of radiosensitivity were calculated using a re-scaled mutual information metric, the information coefficient (IC), a non-linear correlation coefficient that takes values between 1 (perfect association) and 0 (no association) (Fig. 6a).”

—DISCUSSION is excellent. No comments.

Thank you.

452–457: This analysis is OK, but seems very simplistic, statistically speaking. Perhaps some more thought could be put in to a more powerful approach or the signature quality metrics could be explained more

granularly so as to strengthen the reader's confidence in the result. Also, ref to 47 should have author et al. as part of sentence.

We agree. We have re-worked this section by explaining in more granularity the strengths of the signature quality metrics and compared our BRAF gene set to an established gene set that measures ERK activity. The latter is in direct response to reviewer #4's query regarding the use of an ERK pathway gene set and further supports the validity of this new gene set.

The manuscript has been modified as follows:

In **METHODS**, *BRAF gene expression signature*. For each gene from a dataset of total mRNA, we normalized expression values to standard deviations from the median across samples. The most variable genes were identified by calculating the median absolute deviation. The gene list was pruned to 153 on the basis of several gene signature quality metrics including signature gene variability, compactness (as measured by gene autocorrelation) and by the proportion of variance attributable to the first principal component.⁶ Since the signature had both “up” and “down” genes, we calculated, within each sample, the size of the difference of the “up” and “down” genes relative to the variation in each sample to determine the BRAF score. A score that measure that estimates ERK activity on the basis of the BIOCARTA_ERK_PATHWAY (<http://software.broadinstitute.org/gsea/msigdb/index.jsp>) gene set was also computed.

In **RESULTS**, *Phenotypic impact profiling of BRAF variants*. Total mRNA gene expression was assayed using RNAseq. A gene signature composed of the most variable genes was selected to estimate BRAF activity (Supplementary Fig. 2). BRAF scores across all variants had wide and graded variance (Fig. 1e). Importantly, V600E had a high score and the previously characterized modestly hypermorphic mutation G466V had a low score. Genes that comprised the BRAF score significantly overlapped with gene sets that measure epithelial-to-mesenchymal transition ($P=9.05 \times 10^{-26}$; $Q=2.26 \times 10^{-26}$) and KRAS signaling ($P=6.11 \times 10^{-11}$; $Q=7.64 \times 10^{-10}$), consistent with relevant BRAF-related biological pathways (Supplementary Data 1). Moreover, the BRAF score was highly correlated with ERK pathway activity (Pearson $r = 0.783$; Supplementary Fig. 3).

A new supplementary figure that demonstrates the association between BRAF score and ERK activity has also been added.

463–464: Why this formula for tumor volume? Seems very arbitrary. Citation?

Tumor volume based on caliper measurements were calculated by the modified ellipsoidal formula, which is a common method for measuring subcutaneous tumor volumes in mice. The depth is assumed to be equivalent to the shortest of the perpendicular axes due to the frequently difficulty of assessing depth. This approach has been shown to be consistent with other methods of estimating volume (ultrasound, *et cetera*). A citation has been inserted to support this approach.

475: no hyphen between P and value. “of < 0.05” should follow P value. Associated _to_?

Corrected. Now the P value is associated “with”.

487: “ESTIMATE”—citation?

Inserted.

497–498: Each... “species” is a strange analogy. Probably best to delete this sentence. It isn't needed. Shannon index was not invented for ecology or evolutionary biology.

Deleted.

562–567: is it an “association score” or a Coefficient? There isn’t enough information here to interpret what IC is. Please summarize what it is here. Also: “information-based”.

Changed to: “The association between genomic alterations (e.g. *BRAF* mutations) and the radiation response profile was determined using the Information Coefficient (*IC*).⁷⁻⁹ This quantity is obtained by estimating the differential mutual information between the radiation response profile in cells with and without *BRAF* variants. The nominal *P* values for the information-based association metric between the genetic parameters (alterations) and radiation response values were estimated using an empirical permutation test.”

714: is “graded” the best word here?

Changed to: “...and their pathway activity.”

716: “coordinate” is unclear, and is it the majority or the vast majority?

Coordinate has been changed to amino acid position and it was the vast majority.

736: needs more explanation of the “trees”: can you be more specific how to “read”/interpret them? There are lots of kinds of trees; what are these? How does one read the panel nodes, colors, sizes, lines?

Agreed. Changed to: “The clonal evolutionary structure of each tumor is depicted in the form of a rooted tree. The tree with the lowest normalized *log* likelihood value (“best tree”) is shown for representative tumors from each cancer type. Cancer types are organized by color: LUAD (blue), COAD (yellow) and SKCM (red). The root node represents the clonal fraction and branched nodes represent subclones. Node size reflects the number of mutations that constitute the (sub)clone. Arrows indicate the position of the *BRAF* variant(s) in the tree.”

755: t in t-test should be italic throughout

Corrected.

757: locus-specific...single-nucleotide variant

Corrected.

789: Perhaps there is a better way than an empirical permutation test—one that specifies a null hypothesis. If not, clarify how permutation was performed and null that permuted data represents.

The null of the permutation test is that there are no differences between the means of the two groups (cells with v. without an alteration). Since some alterations occur in a small number of samples, this approach is favoured over a parametric test. The permutation test has less assumptions compared to parametric tests (e.g. Gaussianity and independence) and the permutation distribution is the empirical cumulative distribution obtained from the data itself rather than from an idealized distribution. For these reasons, we think that a permutation test is appropriate in this setting.

Figure 1a: y-axis is unclear.

The axis has been made clearer and also incorporates the change from lineage to cancer type.

Very interesting manuscript! —Jeffrey Townsend

We thank the reviewer for their very thorough read of the manuscript and for a multitude of helpful comments.

** References cited*

1. Ryu, D., Joung, J.-G., Kim, N. K. D., Kim, K.-T. & Park, W.-Y. Deciphering intratumor heterogeneity using cancer genome analysis. *Human Genetics* 135, 635–642 (2016).
2. Cannataro, V. L., Gaffney, S. G. & Townsend, J. P. Effect Sizes of Somatic Mutations in Cancer. *J. Natl. Cancer Inst.* 110, 1171–1177 (2018).
3. Pupo, G. M. et al. Clinical significance of intronic variants in BRAF inhibitor resistant melanomas with altered transcript splicing. *Biomark Res* 5, 17 (2017).
4. Xu, Y. et al. Low frequency of BRAF and KRAS mutations in Chinese patients with low-grade serous carcinoma of the ovary. *Diagnostic Pathology* 12, (2017).
5. Dahlman, K. B. et al. BRAF(L597) mutations in melanoma are associated with sensitivity to MEK inhibitors. *Cancer Discov.* 2, 791–797 (2012).
6. Sheen, Y.-S. et al. Prevalence of BRAF and NRAS mutations in cutaneous melanoma patients in Taiwan. *Journal of the Formosan Medical Association* 115, 121–127 (2016).
7. Berger, A. H. et al. High-throughput Phenotyping of Lung Cancer Somatic Mutations. *Cancer Cell* 32, 884 (2017).
8. Cardarella, S. et al. Clinical, Pathologic, and Biologic Features Associated with BRAF Mutations in Non-Small Cell Lung Cancer. *Clinical Cancer Research* 19, 4532–4540 (2013).
9. Hartl, D. L. & Clark, A. G. *Principles of Population Genetics.* (Sinauer Associates Incorporated, 2007).

Reviewer #2 (Remarks to the Author):

This is a well-articulated and coherent paper that makes a coherent case that different BRAF mutation drive different evolutionary trajectories in tumours. They provide well developed and clearly presented experimental data that the BRAF mutations are causal, as well as appropriate analysis of publicly available patient data. All the experiments were well explained in both motivation and significance of the results, and the paper provides a good synthesis of the overall results. The clinical implications are explored with an appropriate preclinical model and support the potential significance of the results.

We thank the reviewer for stating that the manuscript "...is a well-articulated and coherent paper that makes a coherent case that different BRAF mutation drive different evolutionary trajectories in tumours." We also thank the reviewer for recognizing the translational and clinical relevance of this work as suggested by the statement: "the clinical implications are explored with appropriate preclinical model and support the potential significance of the results."

Minor comments:

The TCGA data analysis relies to some degree on similar mutation rates across tumour types, and makes a clock like assumption of mutation that is potentially violated by therapy. It would be good to make mention of this, perhaps in the discussion.

We thank the reviewer for bringing these important topics to our attention. We agree that mutation rates can vary across cancer types and individual cancers. Our results show that if selection causes fixation rapidly, the frequency trajectory of passenger mutations consistently decreases in the low-frequency range of the CCF distribution across a range of neutral mutation rates that have been estimated in several cancers (*i.e.* 10^{-6} - 10^{-9} mutation per base per tumor doubling). The simulator in **Supplementary Software** can be used to adjust the neutral mutation rate. As it relates to the constant mutation rates assumption, mutational signature analysis of subclone mutations strongly support the assumption of a constant mutation rate during subclone evolution.^{10,11} Lastly, we agree that therapy, mainly via cell loss, can significantly alter the neutral mutation rate in a tumor and that violation of the constancy of the neutral mutation rate assumption can impact selection estimates derived from bulk sequencing data. Critically, the TCGA samples that we analysed were treatment naïve. These observations notwithstanding, we agree with the reviewer that it is important to note that our models may not be accurate in the setting of a tumors sampled while on treatment. We have modified the manuscript as follows:

In **DISCUSSION**: "Lastly, therapy, mainly via cell loss, can cause fluctuation in the neutral mutation rate thereby impacting the accuracy of sweep strength estimates for tumors sampled while on therapy and analysed using these methods."

p9 line 189-190 the authors discuss strand. I assume this is meant to be allele? If they wish to avoid the word allele strand is not a good alternative. Copy or template would be preferable, but in my opinion allele or somatic allele would be preferable

Strand has been replaced with allele throughout the text.

It would be preferable if the RNAseq data could be deposited in an appropriate public repository, rather than being available solely by request. If it is to be available only by request it would be ideal if this could be handled by an institutional data access committee.

We have deposited the RNAseq data in GEO. The accession code is GSE133151.

We thank the reviewer for their insights and suggestions.

Reviewer #3 (Remarks to the Author):

The paper by Gopal et al is a very well written description of the role that different BRAF mutations play in terms of their contribution to activation of downstream pathways, tumour growth and clonal evolution. Overall the paper is interesting and the data is beautifully presented. One could argue, however, that it doesn't really tell us something we didn't already know. i.e. that V600E is a potent oncogenic mutation and most other mutations are weaker. Many of the mutations shown have been tested in biochemical/functional assays so we knew this already (1). We also knew that different mutations are associated with different clonal behaviour/subclonality (2), although this paper illustrates this more systematically. Likewise, we know that BRAF mutant tumours (melanomas) have a different genomic landscape compared to tumours without BRAF mutations – this in part is because they can be different melanoma subtypes (the authors appear to group all TCGA subtypes together in their analysis (cutaneous/acral/mucosal) (3). We also already knew that the BRAF locus is amplified in tumours (melanomas) that are BRAF mutant (4) and it is no surprise that amplification is associated with expression differences. The observation that “higher” BRAF variants predict longer progression free survival is also well established (4). Likewise, BRAF mutations occur early (often in nevi) so must, in most cases, proceed amplification of the BRAF locus (5). The authors should be commended for a very detailed job of joining all of these threads together, and writing a paper that is glorious to read, but I don't see the novelty in terms of biological insights.

1. PMID: 30096302

2. PMID: 29748584

3. PMID: 26091043

4. PMID: 29625053

5. PMID: 23690527

We thank the reviewer for stating that “...the paper is interesting and the data is beautifully presented.” We also thank the reviewer for noting that we effectively joined seemingly disparate scientific “threads” and that the manuscript is “glorious to read.”

We address the issue of lack of new “biological insights” in detail below:

This manuscript provides a framework represented in schemas and mathematics that help us *understand* cancer progression. We used *BRAF* as a representative oncogene, but the framework's utility can extend beyond this gene. The models that are presented improve our ability to explain, communicate, explore and predict tumor behaviour and responses to treatments. In this manner, our work provides new insights into previously established biology. For example, there is innovation in the application of population sweep dynamics to clonal selection and response to therapy (see comments from Reviewers #1 & #2). In addition, our models form the basis for the development of a conceptual scaffold that can inform future work. These include the ongoing mapping of the temporal dynamics of clonal sweeps using longitudinal biopsies in PDX (see **DISCUSSION**) and a prospective evaluation of biomarkers determining the role of non-V600E mutations in conferring therapeutic resistance in patients with locally-advanced non-small cell lung cancer receiving chemotherapy and radiation (IRB 07-267: A biomarker-driven strategy to guide the use of chemoradiation in non-small cell lung cancer). Therefore, there is both innovation and pragmatic benefits when biological findings can be translated to testable frameworks. Lastly, as the reviewer notes, our work also places seemingly disparate prior discoveries onto a single canvass, expounds on them in great detail, and conducts rigorous evaluation of new data.

As it relates to biological insights, novelty in a pathway that has been well-worn is challenging. Nevertheless, the manuscript contains several additional advances on prior work and new biological insights. These include the following:

1. Although prior work has identified putative and phenotypically validated cancer drivers in BRAF (e.g. PMID 30096302; PMID: 29625053), there is novelty in: (1) the assignment of these “drivers” into categories of mutations (i.e. ‘hard’ or ‘soft’ drivers in order to estimate strength of selection) using single tumor bulk sequencing data; (2) insights into how individual BRAF alterations can affect the clonal trajectory of the tumor; and (3) the consequences of selection and clonal architecture on therapeutic response to targeted or genotoxic treatments.
2. We agree that prior work has quantified differential selection in well-known drivers, including differences in selection between distinct mutant residues in the same gene like BRAF (e.g. PMID: 29748584). These approaches, however, rely on the frequency at which the mutation is detected in the patient populations as a surrogate marker for the strength of selection. Although these approaches provide some estimates of selection, the vast majority of non-V600E mutations in BRAF occur at very low rates across populations (see Fig. 1A). Our phenotypic profiling of some of these alterations indicated functional significance (as defined by pathway activation and growth; Fig. 1E-1F) despite their low incidence. Due to the rarity of these alterations, however, population-based approaches that use frequency estimates are not sufficiently powered to predict selection in individual patients. A critical advantage of our framework is the ability to estimate selection based on an **individual** patient’s bulk sequencing data. This aspect of our method is now more explicitly described in paragraph 3 of the **DISCUSSION**.
3. We only analysed the BRAF-subtype of cutaneous melanomas (PMID: 26091043). Our comparison was across (SKCM, LUAD, COAD) rather than within cancer types because our focus was on the differences in BRAF trajectories in different tissues of origin.
4. We agree. BRAF amplification in melanomas has been previously studied.^{12,13} Specifically, the functional consequences of copy gains has been well studied in the setting of resistance to targeted therapies. Namely, resistance to BRAFi, MEKi and ERK kinase inhibitors (ERKi) has been shown to be highly correlated with the emergence of copy number gains at the mutated *BRAF* gene locus.¹² However, the functional consequences of copy gains in treatment naïve samples were not known or inherently obvious (see **RESULTS, Copy gain at the BRAF variant locus leads to a hard selective sweep**), neither was the effect of these amplifications on clonal architecture. Our results showed that *BRAF* amplification occurs in treatment naïve samples **and** we demonstrated the functional impact of these alterations on growth and response to therapies. The non-linear and convex (i.e. U-shaped) relationship between *BRAF* copy gain and response to targeted agents that we describe is novel.
5. We agree that based on the presence of *BRAF* V600E in melanocytic nevi (PMID: 23690527), it is probable that V600E mutation preceded amplification for a vast majority of SKCM tumors. However, no similar findings have been reported in LUAD or COAD pre-malignant lesion and the heuristics used to order the genetic events can be applied to other dually altered genes/variants. Therefore, there is novelty in the methods used to order genetic events at a locus.
6. Lastly, our work demonstrates that the vast majority of non-V600 mutant tumors have mainly partial responses to BRAFi/MEKi yet retain functional therapeutic relevance in that they confer resistance to DNA damaging therapies. To our knowledge, we are the first group to demonstrate that BRAF non-V600E mutations can confer resistance to DNA damaging agents. As we indicated in the **DISCUSSION**, clinical data are limited in tumors with non-V600 *BRAF* variants and preclinical data have been inconclusive.¹⁴⁻¹⁶ Several ongoing trials seek to evaluate novel strategies in patients with *BRAF*-mutant NSCLC, including patients with distinct classes of non-V600E mutations.¹⁷ Our results suggest that tumors driven by these variants are more likely to respond to combinatorial therapeutic strategies that include DNA damaging agents. We also suggest that the ordering of therapies (collateral sensitivity) may have significant effects on tumor control in some patients who

can receive both targeted and genotoxic therapy. There is novelty to nominating new therapeutic strategies based on the identity of the *BRAF* mutation and its clonal composition.

We thank the reviewer for challenging us to outline in more explicit terms the added value of this study. We hope that the reviewer finds novelty and utility in both the mathematical framework and some of the new biological insights in this manuscript.

Reviewer #4 (Remarks to the Author):

Gopal and colleagues studied the selection and propagation of BRAF mutation in cancer. They found that V600 mutations hyperactive the ERK signaling pathway and are represented at a higher cancer cell fraction in tumors, as opposed to non-V600 mutations, which are have sub clonal patterns based on CCFs. They also go on to identify evolutionary lineages modeling the evolution of these mutations and correlate these patterns with response to therapy. While the study is interesting and makes some potentially interesting observations there are several limitations.

We thank the reviewer for stating that the “study is interesting” and “makes some potentially interesting observations.”

1. The authors generalize their findings across all BRAF mts yet they only prospectively test V600E or G466V mts. More representatives from each type of BRAF mutations should be tested.

We tested 7 variants representing distinct classes (I-III) of BRAF mutations for *in vivo* growth (Fig. 1f), MEK & ERK activation (Fig. 6b&c), radiotherapeutic resistance (Fig. 6d), and combination targeted and radiotherapy profiling (Fig. 7a). We also determined the optimal variant dose and fitness during selection (targeted drug treatments) for V600E (class I) and G466V (class III) (Fig. 5c). We have now also determined the fitness variant dose landscape of G469A (class II) and R435S (class III); R435S is a novel variant and activates MEK/ERK in part by heterodimerization with CRAF (see Fig. 6b & c). Similar to V600E, a non-linear, convex relationship characterized G469A’s dose-fitness trajectory. Similar to G466V, only higher levels of Dox effected sensitivity to MEKi for R435S. The manuscript has been modified as follows:

In **RESULTS**, *Optimal BRAF variant gene dose and response to BRAFi/MEKi*. “G469A, a high-activity variant, and R435S, a low-activity variant, demonstrated relationships between variant gene dose and drug response similar to V600E and G466V, respectively (Supplementary Fig. 12). These results suggest that both variant identity and dose regulates clonal sweep dynamics, and relatedly, response to targeted agents.”

2. The authors use transcriptional output to score the activity of BRAF mutants. The signature score seems to have been derived from expressing BRAF mts in an immortalized lung cell line. Wouldn't this signature be different if the mutants were expressed in another lineage? Do the mutations rank in a similar fashion if the authors were to use published ERK output signatures to order them?

We used the BEAS2B immortalized lung cell line to test the activity of *BRAF* variants due to the substantial diversity of mutations and the relatively high prevalence of non-V600E mutants in LUAD (compared to SKCM or COAD).¹⁸ We did not test the signature in an immortalized cell line from another tissue of origin. The activity may be dependent on the endogenous activity of RAS and the basal protein levels of ARAF and CRAF for some of the variants, all of which are constant in BEAS2B. ARAF and CRAF levels have not been shown to vary significantly across and with cancer types (see response to Query #6 below).

Yes, the mutations rank in similar order when we used an ERK output signature to order them. We have now incorporated an ERK score to augment the interpretation of the BRAF score. The manuscript has been modified as follows:

In **METHODS**, *BRAF gene expression signature*. “For each gene from a dataset of total mRNA, we normalized expression values to standard deviations from the median across samples. The most variable genes were identified by calculating the median absolute deviation. The gene list was pruned to 153 on the basis of several gene signature quality metrics including signature gene variability, compactness (as measured by gene autocorrelation) and by the proportion of variance attributable to the first principal component.⁶ Since the signature had both “up” and “down” genes, we calculated, within each sample, the size of the difference of the

“up” and “down” genes relative to the variation in each sample to determine the BRAF score. A score that measure that estimates ERK activity on the basis of the BIOCARTA_ERK_PATHWAY (<http://software.broadinstitute.org/gsea/msigdb/index.jsp>) gene set was also computed.”

In **RESULTS, Phenotypic impact profiling of BRAF variants**. “Total mRNA gene expression was assayed using RNAseq. A gene signature composed of the most variable genes was selected to estimate BRAF activity (Supplementary Fig. 2). BRAF scores across all variants had wide and graded variance (Fig. 1e). Importantly, V600E had a high score and the previously characterized modestly hypermorphic mutation G466V had a low score. Genes that comprised the BRAF score significantly overlapped with gene sets that measure epithelial-to-mesenchymal transition ($P=9.05 \times 10^{-26}$; $Q=2.26 \times 10^{-26}$) and KRAS signaling ($P=6.11 \times 10^{-11}$; $Q=7.64 \times 10^{-10}$), consistent with relevant BRAF-related biological pathways (Supplementary Data 1). Moreover, the BRAF score was highly correlated with ERK pathway activity (Pearson $r = 0.783$; Supplementary Fig. 3).”

Supplementary Fig. 3 has been added to demonstrate the similarity between BRAF score and ERK activity.

3. The authors should compare the clonal trajectory of the V600E mt form lung, colon and melanomas and determine if these are always clonal regardless of lineage. The way that the text is written makes is difficult to conclude if the clonal pattern of this mutation is due to its biochemical properties or due to lineage-specific factors.

We agree that this should be more explicit. We have modified the manuscript to reflect our estimates of clonality for tumors with V600E mutations as follows:

In **RESULTS, BRAF variant fitness and tumor clonal architecture**: “Estimates of the ratios of clonal BRAF V600E variants in LUAD, COAD and SKCM tumors were 0.33, 0.52 and 0.76, respectively.”

4. The authors generated isogenic cell lines of 35 variants and correlated high output scores with faster tumor formation. Did all variants have the same protein expression level?

The 34 constructs expressing variants and wild type resulted in similar BRAF gene expression values. The \log_2 RNAseq BRAF expression values from variant-expressing cells ranged from 10.9-15.8 (figure for review). The vector control value was 7.5, indicating a 1.5 to 2.1-fold increase in gene expression in BEAS2B. Importantly, BRAF gene expression values were not correlated with BRAF or ERK scores ($R^2 = 0.003$ & 0.03 , respectively). We also measured the protein expression of some variants (figure for review) and found some variation in protein expression, although this did not correlate significantly with BRAF activity. Together, these results indicated that heterologous BRAF gene and protein expression levels are similar across the transfected cell lines and do not alone account for the variability in BRAF/ERK-induced activity.

5. The authors show that *BRAF* V600E amp increases tumor growth rate in the absence of therapy. This is not consistent with data showing that *BRAF* amp is at a fitness disadvantage in absence of treatment. This needs to be addressed.

Yes, higher expression of *BRAF* V600E under the moderately expressing EF-1 α promoter compared to lower expressing PGK promoter increases the growth rate of BEAS2B cells (Supplementary Fig. 7). We are also keenly aware of the results that indicate that *BRAF* amplification while on therapy can result in a fitness disadvantage upon withdrawal of treatments.^{19,20} These two results are not mutually exclusive and both are concordant with our model. We posited that the optimal fitness of cells, including our immortalized lung cell line, are dependent on a precise level of *BRAF*-MEK-ERK pathway activation, such that too little, in response to lower levels of gene expression (see inducible system) or pathway inhibition, or too much, in response to significant overexpression or drug withdrawal leading to elevated flux through the pathway are both deleterious to cell proliferation. These contentions also form the basis of (and explain) the observed non-linear and convex (*i.e.* U-shaped) relationship between *BRAF* copy gain and response to targeted agents (Fig. 5D). The observations in Supplementary Fig. 7 are a matter of extent of expression due the exigent strategy of using the particular constitutive promoters for *in vivo* growth. To illustrate these principals further, we measured the relative growth of *BRAF* (V600E, G469A, and G466V)-expressing BEAS2B cells under the control of the tunable Dox inducible promoter *in vitro*. We observed induced senescence of BEAS2B cells only at the highest levels of expression of the most hyperactive allele, V600E (figure for review). Of note and relatedly, the V600E curve is non-linear and concave, which is, expectedly, a horizontal reflective approximation of the U-shaped curve in Fig. 5D.

6. What does a 'moderately hypermorphic variant' mean? G466V has been described on several occasions to be a low activity mutant. This mutation transforms by dimerizing with WT RAFs. Why are the authors surprised that induced expression of G466V doesn't activate ERK in the same dose dependent manner as V600E? Wouldn't the effect of G466V (regardless of how much of this is expressed) be limited by the level of endogenous A/B/CRAF expression (since the later are required to dimerize and provide the necessary kinase activity to stimulate the ERK pathway)? This experiment sheds little light into the effects of amplification on fitness. The authors should compare the induced expression of V600E to the induced expression of other high activity mutants.

Moderately hypermorphic means low activity mutant. We agree that this can be confusing. Therefore, we have eliminated the qualification of hypermorphic throughout the manuscript and instead now use the previously established nomenclature of high-, intermediate- or low-activity mutations.²¹

We also concur that the mutation leads to pathway activation via dimerization with wild type RAFs. We sought to determine whether “amplification” (effected here by increasing heterologous gene expression) of this allele could have functional consequences. We showed that increasing gene expression causes dose-dependent increases in ERK activity (Supplementary Fig. 6) resulting in increased growth but not resistance to MEKi treatments. The latter was not observed at highest levels of gene expression (Fig. 5). We believe these results illuminate the functional consequences of the potential amplification of this allele. Moreover, as we now demonstrate, despite prior claims that amplification at the *BRAF* locus can result in resistance to targeted therapy, this effect appears to be specific to only some *BRAF* variants, namely high activity mutants (*e.g.* V600E & G469A; below).¹²

We are not aware of data that indicates differences in RAF1 or ARAF basal protein expression across groups of tumors. Data from TCGA indicates that levels do not vary substantially across and with cancer types (figures for review).

Per the reviewer's query, we have also compared the induced expression of another high activity mutants with V600E. We show that similar to V600E, G469A mutant cells strongly activate MEK and ERK1/2 in a dose-dependent manner and demonstrate a non-linear and convex (*i.e.* U-shaped) relationship between *BRAF* levels and response to targeted agent treatments (Supplementary Fig. 6 & Supplementary Fig. 12).

7. They show that BRAF amp is sensitive to RAF/MEK inhibitor combination. What is the level of amplification at which this effect was observed? It seems that at higher levels of dox induced BRAF expression the tumors grow while on this treatment (Fig 5d). What is the rationale for testing the RAF/MEK inhibitor combination in non V600E mutant settings? These should not respond to the RAF inhibitor. Did the authors use a dimer inhibitor?

As discussed in Query #5, we showed that the optimal fitness of cells (and therefore the likelihood of response to targeted agents), including our immortalized lung cell line, are dependent on the level of BRAF-MEK-ERK pathway activation, such that too much, in response to significant overexpression via amplification leading to elevated flux through the pathway can diminish the likelihood of response to targeted therapies (resistance).

We mainly used RAFi for V600 mutants and MEKi for non-V600 mutants. The singular instance of using a RAFi+MEKi combination in non-V600 mutant experiment in the manuscript was in Fig. 5b when testing the G469A PDX. The rationale was to subject the mice in all cohorts to the same *in vivo* drug combination to minimize experimental cohort differences to allow for a precise measurement of tumor growth kinetics. Moreover, there is evidence that suggests that the combination of Dabrafenib and Trametinib in cells with G469 mutations leads to lower p-ERK levels than either drug alone.¹⁶

Minor-

1. The experimental work should be described better, particularly that describing the effect of amplification, RAF/MEK combination and DNA damage inducing agents. The authors spend a lot of time describing computational analysis, and this comes at the price of an appropriate description of the experimental work.

We have modified the **METHODS** in several experimental sections in order to better describe the experimental work.

2. Some of the ideas in the manuscript could be conveyed in a simpler and more direct manner to the reader.

Due to the trans-disciplinary content of the manuscript and the audience of this journal, we felt it was important to provide background to contextualize some of the topics. This may have contributed to verbosity in some sections. By some of the above annotated revisions, we have made additional attempts to be more direct and clearer.

We thank the reviewer for their insights and suggestions.

REFERENCES

1. Ryu, D., Joung, J.G., Kim, N.K.D., Kim, K.T. & Park, W.Y. Deciphering intratumor heterogeneity using cancer genome analysis. *Human Genetics* **135**, 635-642 (2016).
2. Cannataro, V.L., Gaffney, S.G. & Townsend, J.P. Effect Sizes of Somatic Mutations in Cancer. *J Natl Cancer Inst* **110**, 1171-1177 (2018).
3. Temko, D., Tomlinson, I.P.M., Severini, S., Schuster-Bockler, B. & Graham, T.A. The effects of mutational processes and selection on driver mutations across cancer types. *Nat Commun* **9**, 1857 (2018).
4. Neuhauser, C. & Krone, S.M. The genealogy of samples in models with selection. *Genetics* **145**, 519-34 (1997).
5. Smith, J.M. What use is sex? *J Theor Biol* **30**, 319-35 (1971).
6. Dhawan, A. *et al.* sigQC: A procedural approach for standardising the evaluation of gene signatures. *bioRxiv* (2017).
7. Abazeed, M.E. *et al.* Integrative radiogenomic profiling of squamous cell lung cancer. *Cancer Res* **73**, 6289-98 (2013).
8. Joe, H. Relative Entropy Measures of Multivariate Dependence. *Journal of the American Statistical Association* **84**, 157-164 (1989).
9. Linfoot, E.H. An informational measure of correlation. *Information and Control* **1**, 85-89 (1957).
10. Williams, M.J. *et al.* Quantification of subclonal selection in cancer from bulk sequencing data. *Nat Genet* **50**, 895-903 (2018).
11. Alexandrov, L.B. *et al.* Signatures of mutational processes in human cancer. *Nature* **500**, 415-21 (2013).
12. Xue, Y. *et al.* An approach to suppress the evolution of resistance in BRAF(V600E)-mutant cancer. *Nat Med* **23**, 929-937 (2017).
13. Genomic Classification of Cutaneous Melanoma. *Cell* **161**, 1681-96 (2015).
14. Lin, L. *et al.* Mapping the molecular determinants of BRAF oncogene dependence in human lung cancer. *Proc Natl Acad Sci U S A* **111**, E748-57 (2014).
15. Blay, J.Y. *et al.* Vemurafenib (VM) in non-melanoma V600 and non-V600 BRAF mutated cancers: first results of the ACSE trial. *Annals of Oncology* **27**, 55PD-55PD (2016).
16. Noeparast, A. *et al.* Non-V600 BRAF mutations recurrently found in lung cancer predict sensitivity to the combination of Trametinib and Dabrafenib. *Oncotarget* **8**, 60094-60108 (2017).
17. Odogwu, L. *et al.* FDA Approval Summary: Dabrafenib and Trametinib for the Treatment of Metastatic Non-Small Cell Lung Cancers Harboring BRAF V600E Mutations. *Oncologist* **23**, 740-745 (2018).
18. Paik, P.K. *et al.* Clinical characteristics of patients with lung adenocarcinomas harboring BRAF mutations. *J Clin Oncol* **29**, 2046-51 (2011).
19. Sale, M.J. *et al.* MEK1/2 inhibitor withdrawal reverses acquired resistance driven by BRAF(V600E) amplification whereas KRAS(G13D) amplification promotes EMT-chemoresistance. *Nat Commun* **10**, 2030 (2019).
20. Petti, C. *et al.* Coexpression of NRASQ61R and BRAFV600E in human melanoma cells activates senescence and increases susceptibility to cell-mediated cytotoxicity. *Cancer Res* **66**, 6503-11 (2006).
21. Yao, Z. *et al.* Tumours with class 3 BRAF mutants are sensitive to the inhibition of activated RAS. *Nature* **548**, 234-238 (2017).

REVIEWERS' COMMENTS:

Reviewer #2 (Remarks to the Author):

The revised manuscript is greatly improved in clarity and all my concerns have been fully addressed.

Reviewer #3 (Remarks to the Author):

The authors have done a significant amount of work to revise their manuscript. What I like about this paper is that it is so clear and well structured and thoughtful but as conceded by the authors (page 10 of the responses) is less forthcoming with biological insights.

The question is whether this work represents a significant methodological advance? In this regard I do think that they provide some clarity but have largely worded around many of my comments rather than answering them directly.

I am finding it hard to assess the suitability of this paper for Nature Communications but note that one of the other reviewers who is an expert in this aspect of the paper is positive.

Reviewer #4 (Remarks to the Author):

The authors have addressed the comments raised in my initial review. The revised manuscript is significantly improved and I have no further concerns.

Reviewer #5 (Remarks to the Author):

I have checked the responses to Reviewer 1's comments and suggestions and found all issues to have been satisfactorily addressed.